# Modifying electron transfer between photoredox and organocatalytic units via framework interpenetration for β-carbonyl functionalization

Zhengqiang Xia[1], Cheng He[1], Xiaoge Wang[1] & Chunying Duan[1,2,3]

Modifying electron transfer pathways is essential to controlling the regioselectivity of heterogeneous photochemical transformations relevant to saturated carbonyls, due to fixed catalytic sites. Here we show that the interpenetration of metal–organic frameworks that contain both photoredox and asymmetric catalytic units can adjust the separations and electron transfer process between them. The enforced close proximity between two active sites via framework interpenetration accelerates the electron transfer between the oxidized photosensitizer and enamine intermediate, enabling the generation of $5\pi e^-$ β-enaminyl radicals before the intermediates couple with other active species, achieving β-functionalized carbonyl products. The enriched benzoate and iminium groups in the catalysts provide a suitable Lewis-acid/base environment to stabilize the active radicals, allowing the protocol described to advance the β-functionalization of saturated cyclic ketones with aryl ketones to deliver $\gamma$-hydroxyketone motifs. The homochiral environment of the pores within the recyclable frameworks provides additional spatial constraints to enhance the regioselectivity and enantioselectivity.

[1] State Key Laboratory of Fine Chemicals, Dalian University of Technology, Dalian 116024, China. [2] College of Zhang Dayu, Dalian University of Technology, Dalian 116024, China. [3] Collaborative Innovation Center of Chemical Science and Engineering, Tianjin 300071, China. Correspondence and requests for materials should be addressed to C.D. (email: cyduan@dlut.edu.cn)

Catalytic synthesis methods that work under ambient atmosphere with benign reaction environments and clean energy have been a major goal in synthetic chemistry[1]. Of the essential structural motifs that are frequently found in pharmaceutical, material, agrochemical, and fine chemicals, the carbonyls play a pivotal role as powerful building blocks in broad areas of synthetic organic chemistry[2–4]. The functionalization of carbonyls, one of the most fundamental transformation in organic synthesis, has evolved to a range of widely used organic reactions and synthetic protocols[5]. While direct α-carbonyl substitution is flourishing because the α-methylene position of the carbonyl group can be readily functionalized with diverse nucleophiles and electrophiles via enolate or enamine chemistry[6–8], the direct β-activation of saturated carbonyls has proven to be a more cumbersome and challenging task owing to the typically unreactive β-C(sp$^3$)-H bonds and competitive α-substitution or other reactions[9–11]. Most primary β-carbonyl substitution is still achieved indirectly by the nucleophilic addition of α, β-unsaturated carbonyls[12, 13], despite a few examples of direct β-activation having been reported by enamine oxidation or palladium catalysis[14, 15].

On the other hand, light-promoted photocatalysis, a mild and green chemical synthesis approach, furnishes powerful molecular activation modes to enable bond constructions that are elusive or impossible via established protocols[16, 17], undoubtedly granting a new opportunity for the direct β-position bonding of saturated carbonyls. Recent investigation has also suggested that the combination of photoredox catalysis with enamine organocatalysis is a promising strategy for the β-functionalization of saturated carbonyls through a 5πe$^-$ carbonyl activation mode[18]. It is postulated that the clean energy activation mode applies to a wide range of β-carbonyl functionalization transformations and is also compliable with asymmetric catalysis; meanwhile, the long-life and high-energy radical in such homogeneous system requires the co-existence of several types of catalysts and adducts to enhance its stability and ensure effective coupling reactions.

However, in heterogeneous systems, owing the fixed catalytic sites, the modification and optimization of the rapid electron transfer between the two catalytic cycles for regioselectivity control should be given important consideration during the design of the catalysts. In this case, the major challenge goes beyond the incorporation of suitable photoredox centers and organocatalytic centers within the catalytic systems and includes the precise regulation of the electron transfer pathway between the active sites. New strategies wherein the location of all the catalytic requisites can be modified in an easy-to-control model with adjustable electron states and interactional patterns appear imminent to achieve photocatalytic transformation.

Metal–organic frameworks (MOFs) are fascinating hybrid solids with infinitely ordered networks consisting of organic bridging ligands and inorganic nodes[19, 20]. The tenability and flexibility of the polymers granted by the diversity of their building blocks allow the incorporation of photoactive organic dyes and the necessary adducts into a single framework, which represents a new approach to heterogenizing several photo-catalytic conversions[21]. The regular distribution of catalytic sites within the confined micro-environment benefits the fixation and stabilization of the active radicals formed under irradiation to overcome restrictions of homogeneous processes[22]. The long-range ordered structure of a MOF-based crystalline material contains precisely located catalytic centers, which provides an excellent platform to study photon capture and electron transport and delivery[23, 24]. As the integration of both photocatalysis and asymmetric organocatalysis into a single MOF has been considered a promising approach to the α-functionalization of saturated carbonyls[25], we believe the modification of electron transfer between the two active sites within the frameworks via framework interpenetration is a potential strategy for the β-functionalization of saturated ketones and aldehydes.

Here we show that the interpenetration of the polymers allowed the appropriate adjustment of the separation and inter-actions between the photoredox and organocatalytic sites. With

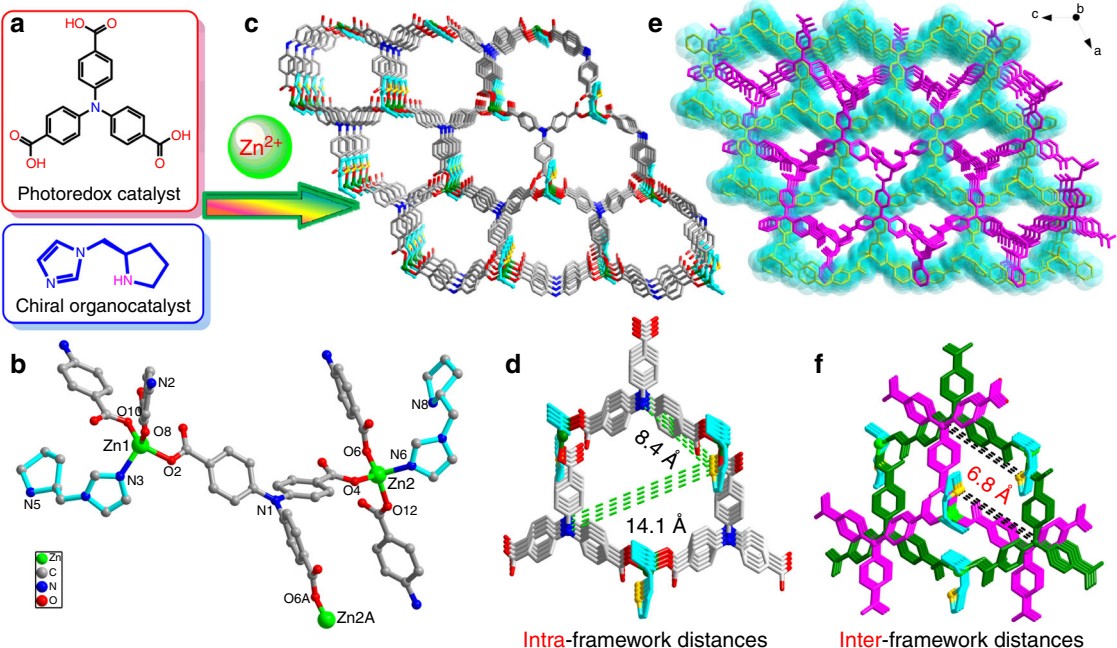

**Fig. 1** The assembly and crystal structure of **InP**-1. The components (**a**) and coordination environments of the Zn(II) ions and **NTB$^{3-}$** ligand (**b**) in **InP**-1; Symmetry code: A, x + 1/2, y-1/2, z. 3D structures of the isolated frameworks viewed along the b-axis, before **c** and after **e** the twofold interpenetration. The intra-(**d**) and inter-(**f**) framework separations between the catalytic sites. Atoms in the pyrrolidine rings of L-**PYI** are marked in yellow and turquoise. H-atoms and solvent molecules are omitted for clarity

the closer interframework distances, the electron transfer between the two active sites is accelerated, such that the special $5\pi e^-$ β-enaminyl radical is more likely to be quickly generated from the enamine intermediate to achieve β-functionalized carbonyl products before the enamine intermediate is coupled with other active species. As a heterogeneous photocatalysis for the direct β-functionalization of saturated carbonyls, the ability to recover and reuse the photocatalysts makes our system interesting in terms of the practical applications of the photochemical transformation. Additionally, the enriched benzoate and imine groups in the polymers create a suitable Lewis acidic/basic environment to replace the complex catalytic adducts in a homogeneous system to further stabilize long-life and high-energy radicals, which makes the catalytic system simpler, milder and more environmentally friendly. The matched redox potentials of the interpenetrated polymers promoting this catalytic protocol are further advanced to allow the β-functionalization of the other hard-to-reduce saturated carbonyls.

## Results

**Synthesis and characterization of the interpenetrated MOF.** Solvothermal reaction of *L*-pyrrolidin-2-ylimidazole (*L*-**PYI**)[26], 4,4′,4″-nitrilotrisbenzoicacid (**H₃NTB**), and Zn(NO₃)₂·6H₂O afforded a hybrid crystalline solid, **InP**-1 (Fig. 1a, Supplementary Data 1). Elemental analysis and powder X-ray diffraction demonstrated the potential phase purity of the bulk sample (Supplementary Fig. 1). Single-crystal X-ray diffraction revealed

the formation of the twofold interpenetrated coordination polymer and the crystallization into the chiral space group $C_2$. Within each of the identically isolated chiral framework, the Zn(II) ion was coordinated in a chiral tetrahedral geometry with three carboxylate oxygen atoms from three different **NTB³⁻** ligands and one nitrogen atom from the **PYI** coligand. The propeller-like **NTB³⁻** ligand rotated the arms with special angles to meet the requirements of the Zn(II)-tetrahedron geometries (Fig. 1b) and linked three zinc ions through the carboxylate groups to generate a 3D framework containing 1D chiral channels with a cross section of $16.7 \times 17.0\,\text{Å}^2$ (Fig. 1c). Two symmetric related frameworks were interpenetrated to form the crystalline solid, with the opening of the channels being reduced to $14.3 \times 9.4\,\text{Å}^2$ (Fig. 1e, Supplementary Figs. 9 and 10). Chiral **PYI** catalytic sites were regularly distributed on the inner surfaces of the channels with the exposed active amine groups (N−H pyrrolidine) working as accessible catalytic sites to active aldehydes or ketones.

Most importantly, the interpenetration of the two frameworks enforced the close proximity between the photoredox sites (**NTB³⁻**) and the asymmetric organocatalytic sites (**PYI**), enabling the shortest intraframework N···N separation of 8.4 Å between the pyrrolidine and triphenylamine N atoms in each non-interpenetrating framework aforementioned shorten to 6.8 Å in the interpenetrated **InP**-1 (Fig. 1d, f). The formed structural motif will benefit the acceleration of the electron transfer between the enamine intermediate and the photosensitizer. This acceleration promotes the generation of the $5\pi e^-$ β-enaminyl radical, a key step for the achievement of the β-activation of carbonyl, being

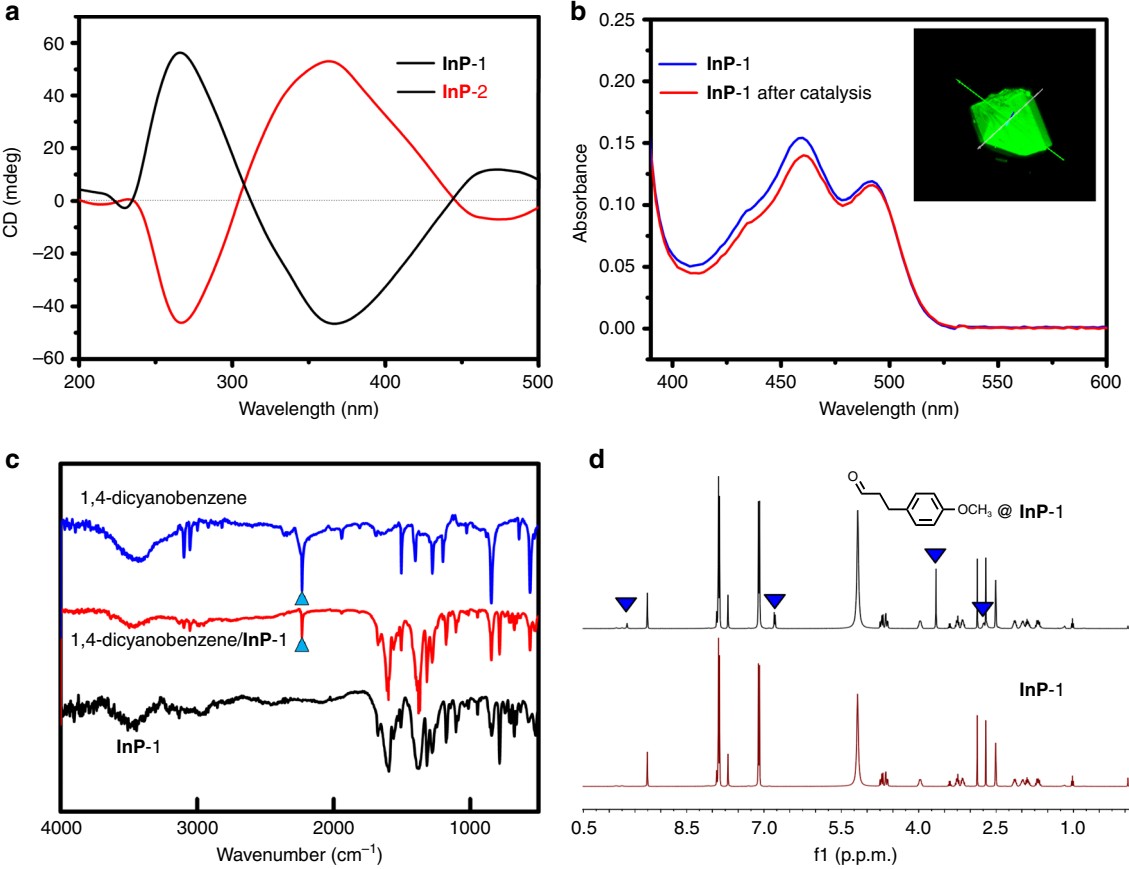

**Fig. 2** The chirality and porosity characterization of **InP**-1. **a** CD spectra of bulk crystalline solids of **InP**-1 and **InP**-2 showing the opposite Cotton effects of the two polymers. **b** UV–vis spectra of 2′,7′-dichloro fluorescein dye released from **InP**-1 and **InP**-1 after catalysis. Inset: the 3D reconstruction of **InP**-1 soaked with 2′,7′-dichlorofluorescein dye. **c** IR spectra of 1,4-dicyanobenzene, **InP**-1, and **InP**-1 with absorbed 1,4-dicyanobenzene. The light blue triangles show the characteristic $\nu^{st}_{C\equiv N}$ signals of 1,4-dicyanobenzene. **d** ¹H NMR spectra of **InP**-1 and **InP**-1 with adsorbed 3-(4-methoxyphenyl) propionaldehyde dissolved in DMSO-$d_6$/DCl. Peaks marked with *blue triangles* represent the signals of 3-(4-methoxyphenyl) propionaldehyde

more rapid than the direct coupling reaction of the enamine intermediate with other active species. Additionally, the free migration of protons around the uncoordinated nitrogen or oxygen atoms of the ligands and the chiral organocatalysts allows the enriched Lewis acid and Lewis base to possibly act as catalytic adducts to stabilize high-energy radicals. This structural motif grants a possibility of replacing the complex catalytic adducts of the homogeneous system[18], making the reaction conditions simpler, cleaner and more environmentally friendly, and allowing separation, and recovery in potential practical applications.

The chirality of the interpenetrated frameworks was identified by the chiral space group it crystallized into and confirmed by the circular dichroism (CD) spectrum, with a positive Cotton effect centered at ~263 nm and a negative Cotton effect centered at ~366 nm (Fig. 2a and Supplementary Table 1). The porosity of the polymer was determined by the PLATON software[27] with 18.8% of the whole crystal volume calculated. To evaluate the accessibility and stability of the channels to the bulky organic substrates in the solution environment, a dye-uptake assay, an increasingly common method was performed before the MOF-based catalyst was used in catalytic investigations[28, 29]. In the general procedure, the as-synthesized crystalline solid InP-1 was first soaked overnight in a methanol solution containing 2′,7′-dichlorofluorescein. The resulting crystalline solid was filtered and washed with methanol several times to remove the potential dye molecules adsorbed on the external surfaces of the sample. The dye-absorbed InP-1 was then dissociated by concentrated hydrochloric acid, and the resultant clear solution was diluted to 10 mL and adjusted to a pH of 1.5. The quantum dye-uptake was determined as 12.5% of the solid weight by ultraviolet–visible (UV–vis) spectroscopy (Supplementary Fig. 6). Confocal laser scanning microscopy of the dye-adsorbed crystal showed the successful uptake of the dye molecules within the channels of the polymer (Fig. 2b and Supplementary Fig. 7)[30, 31], which revealed the good stability of the well-maintained pores in solution and the possibility of the substrates diffusing inside the channels.

To further confirm the capability of InP-1 to transport substrate molecules through the open channels, the InP-1 was immersed in solutions of $CH_2Cl_2$ containing the possible substrates 1,4-dicyanobenzene or 3-(4-methoxyphenyl)propional-dehyde. Infrared (IR) spectrum of InP-1 with adsorbed 1,4-dicyanobenzene showed the characteristic signals of $\nu^{st}_{C\equiv N}$ at 2231 cm$^{-1}$ (Fig. 2c), and the $^1H$ NMR of InP-1 with adsorbed substrates revealed that InP-1 could adsorb ~1.3 equiv of 1,4-dicyanobenzene or 1 equiv of 3-(4-methoxyphenyl)propionalde-hyde per triphenylamine moiety (Fig. 2d and Supplementary Fig. 20). All these results demonstrated that the pores of the MOFs are accessible for smooth substrate diffusion and suggested that InP-1 was suitable for heterogeneous catalysis. Furthermore, because the InP-1 was evolved from two of the same non-interpenetrating frameworks via interpenetration, this transforma-tion provides a chance of adjusting the pore environment and at the same time enhancing the density of triphenylamine groups around the pores.

**Photocatalytic β-functionalization of saturated aldehydes.** The β-arylation reaction was initially examined by employing 1,4-dicyanobenzene and propionaldehyde as the coupling partners, along with a 20 watt fluorescent lamp as the light source. A 5 mol% loading of InP-1 gives an 87% conversion to the β-functio-nalization under irradiation without the formation of any $\alpha$-functionalization adduct (entry 1, Table 1). Compared with the complex reaction conditions (multiple additives: base, water, acid, photosensitizer, catalyst, and 1,3-dimethyltetrahydropyrimidin-2 (1$H$)-one (DMPU) solvent) in the recently reported

homogeneous β-functionalization of carbonyls[18], our catalytic system afforded efficient and selective transformation in the presence of only the organic base 1,4-diazabicyclo[2.2.2]octane (DABCO) and the designed MOF-based catalysts in N,N-dime-thylformamide (DMF) solution, which made our reaction con-ditions simpler, milder, and more manageable. Moreover, the heterogenization of the organic photoredox catalyst H₃NTB into the frameworks replaced the noble-metal Ir(ppy)₃ in the homo-geneous systems, which not only prevented heavy metal pollution on final reaction products but also saved the costs of synthesis and post-processing in mass production, rendering our system less costly and more environmentally friendly.

The InP-1 solids were easily isolated from the reaction suspension by simple filtration. The removal of InP-1 by filtration after 12 h of irradiation shut down the reaction directly, with the filtrate affording only 4% additional conversion over another 36 h of irradiation (Fig. 3a). Washing the filtered residues with DMF enabled their direct reuse in a new round of reactions. After three rounds of reuse, InP-1 solids exhibited a moderate loss of reactivity (87~82% conversion; Fig. 3b and Supplementary Table 5). The PXRD pattern of the InP-1 filtered from the reaction mixture matched well with the simulated one based on the single-crystal simulation, suggested the maintenance of the MOF framework (Supplementary Fig. 1). Additionally, further confirming the maintenance of the pores of the catalysts, dye-uptake studies for the recycled InP-1 exhibited an 11.7% dye-uptake corresponding to the framework weight. The comparable adsorption ability before and after the catalytic reaction confirmed the chemical stability of the well-preserved pores (Fig. 2b). The results suggested the successful execution of our structural design and demonstrated that InP-1 was a hetero-geneous catalyst with advantages of easy separation and recovery[32] that utilizes mild and more environmentally friendly reaction conditions and has potential practical applications.

Control experiments showed that no transformation occurred in the presence of only the 5 mol% H₃NTB, the 5 mol% PYI or the 5 equiv of DABCO base. Almost negligible conversion was observed when the reaction was performed in the dark (entries 1–5, Supplementary Table 4). To directly compare the MOF-catalyzed reaction with the standard homogeneous conditions[18], the same adducts and solvent (DABCO, HOAc, water, and DMPU) in equivalent amounts were added to catalyze the transformation with free catalysts, but no β-arylation product was detected. This result suggested that the interpenetrated MOF catalysts played an indispensable role in our heterogeneous system (entry 6, Supplementary Table 4). Considering the catalytic functions of H₃NTB and PYI, 5 mol% H₃NTB and PYI were loaded as the homogeneous catalysts to catalyze the reaction under optimal conditions. A low conversion of 11% was observed because the arms of H₃NTB rotated quickly in the solvent and gave very poor light absorption (entry 8, Supple-mentary Table 4)[33–35]. Even when adding extra zinc salts to coordinate with the H₃NTB and thereby hinder the rotation, only a small increase to 26% conversion could be obtained.

A further comparison of the catalytic efficiency was based on an interpenetrated achiral coordinated polymer (InP-3) with almost the same structure, except the chiral organocatalytic site was substituted by a pyridine moiety (Fig. 3c, Supplementary Data 3 and Supplementary Fig. 11). As expected, the loading of 5 mol% InP-3 under the same reaction conditions as InP-1 yielded negligible conversion. The two structurally similar compounds have almost the same ground redox potential ($E_{1/2} \approx 0.83$ V vs. SCE) in the solid state and quite similar absorption and emission spectra, and the excited state redox potentials of −2.16 V and −1.97 V of InP-1 and InP-3, respectively (Supplementary Figs. 4 and 5)[36], are both more negative than those of organomatallic Ir

**Table 1 Conversion and enantiomeric excess (ee) of the photocatalytic β-arylation of saturated aldehydes with InP-1 and InP-2 as bifunctional heterogeneous catalysts[a]**

| Entry | Substrate[b] | Molecular size[b] | Conversion/%[c] | ee/%[d] |
|-------|------------|------------------|-----------------|---------|
| 1 |  |  4.4 Å, 3.1 Å | 87 (84) | — |
| 2 |  |  6.9 Å, 3.1 Å | 84 (82) | 28 (–30) |
| 3 |  |  10.7 Å, 3.1 Å | 75 (77) | 28 (–29) |
| 4 |  |  11.6 Å, 4.3 Å | 69 (65) | 29 (–29) |
| 5 |  |  8.6 Å, 4.3 Å | 78 (80) | 49 (–45) |
| 6 |  |  10.8 Å, 4.3 Å | 72 (68) | 52 (–55) |
| 7 |  |  14.9 Å, 12.9 Å | <5 | — |

[a]Reaction conditions: 1,4-dicyanobenzene (1.0 mmol), aldehydes (1.4 mmol), **InP**-1/**InP**-2 (5 mol%, based on **H₃NTB**), DABCO (5 mmol), DMF (3 mL), in N₂, 20 watt fluorescent lamp, 48 h; The values in parentheses represent the results yielded by the **InP**-2 catalyst
[b]The assumed structures and molecular sizes were calculated via the Chem3D program
[c]The conversions were determined by GC analysis using biphenyl as an internal standard
[d]The ee values were determined by HPLC analysis

(ppy)₃ ($E_{1/2}{}^{ox}$ = −1.73 V)[37, 38] and the substrate 1,4-dicyanobenzene ($E_{1/2}{}^{red}$ = −1.61 V)[39]. The difference in the catalytic efficiency is attributed to the absence of synergistically organocatalytic groups that used to activate the substrate. This postulation was confirmed by the conversion of the aforementioned catalytic system increasing to 58% upon the excess addition of the organocatalytic L-**PYI** to the reaction mixture.

From a mechanistic point of view, in our heterogeneous system, the interpenetrated coordinated polymer **InP**-1 adsorbed

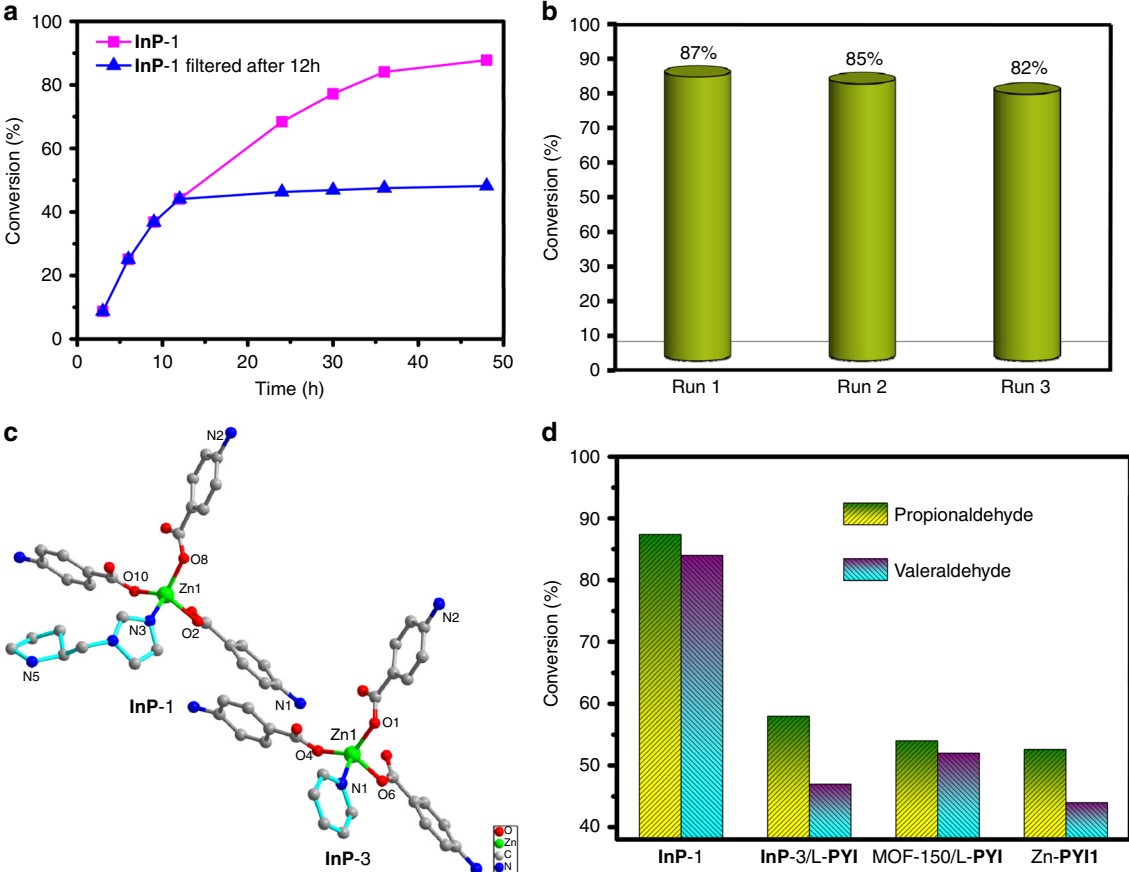

**Fig. 3** The photocatalysis results and control experiments. **a** Catalytic traces of the β-arylation of propionaldehyde performed by **InP**-1 and **InP**-1 filtered after 12 h under the optimal conditions. **b** Recycling catalytic experiments of the β-arylation of propionaldehyde catalyzed by **InP**-1 under the optimum conditions. **c** The coordination environments of Zn(II) ions in **InP**-1 and **InP**-3. **d** Conversion levels of the β-arylation reaction between 1,4-dicyanobenzene and saturated aldehydes using different catalysts (5 mol%)

1,4-dicyanobenzene within its channels, and this substrate quenched the excited state of the heterogeneous catalyst to afford the radical anion and oxidized **InP**-1⁺. The aldehyde substrates readily diffuse inside the channels and are activated by the chiral amine catalyst **PYI** moiety to potentially form the normal chiral enamine intermediate. The close proximity between the photo-redox catalytic site and the organocatalytic site increased the opportunity for the enamine intermediate to be quickly oxidized by **InP**-1⁺ through interframework electron transfer to generate the corresponding enaminyl radical cation, rather than direct coupling with the arene radical anion or other species that created bonds at the α-carbonyl position. The high-energy enaminyl radical cations were favorable to increase the acidity of the allylic C–H bonds and further promote the corresponding deprotonation at the β-position of carbonyl, eventually completing the critical 5πe⁻ activation mode. The newly formed 5πe⁻ β-enaminyl radical quickly coupled with the 1,4-dicyanobenzene radical anion to yield the cyclohexadienyl anion, which further underwent hydrolysis to give the β-arylation product and return the **PYI** (Fig. 4 and Supplementary Fig. 15). This prospective is confirmed by the luminescent quenching of the interpenetrated polymer upon the addition of 50 μM 1,4-dicyanobenzene substrate, with Stern-Volmer constants of the constructed polymers **InP**-1 and **InP**-3 of $3.56 \times 10^3$ and $3.83 \times 10^3$ M⁻¹, respectively. The corresponding fluorescence lifetime decreased from 4.62 to 3.25 ns, which revealed a classical photo-induced electron transfer process from the excited state of the catalyst to the substrate and a fast charge transfer rate (Supplementary

Figs. 21–24)[40]. The IR spectrum of **InP**-1 immersed in a DMF solution of propionaldehyde revealed a C=O stretching vibration at 1724 cm⁻¹. The redshift from 1736 cm⁻¹ (free propionaldehyde) suggested the adsorption and activation of propionaldehyde in the pores of the polymer (Supplementary Fig. 16). These affinities implied the probability of the formation of the key enamine intermediate. Additionally, the fact that the relevant shift was not observed in IR spectrum of **InP**-3-adsorbed propionaldehyde confirmed the inactivity of **InP**-3 under the catalytic processes (Supplementary Fig. 25).

The unique crystalline characteristic of the MOFs yields the opportunity to learn their complex internal structure by single-crystal X-ray diffraction technique. In addition, the precise determination of the crystal structure of the framework-encapsulated substrate molecules provides more chemical information about the host-guest interaction patterns, paths, and sites distribution, which allows us to better understand the mysterious activation and formation processes of the catalytic intermediate and has become a promising approach for catalytic studies[41, 42]. Thus, the **InP**-1 crystals were soaked in a solution containing substrates for 12 h, and fortunately the quality of the propionaldehyde-impregnated **InP**-1 crystals (denoted **InP**-4) was suitable for X-ray crystallographic analysis (Supplementary Data 4 and Supplementary Figs. 12 and 18). The structural analysis of **InP**-4 revealed that the space group and unit cell sizes were almost the same as those of the original crystal. As shown in Fig. 5a, the propionaldehyde substrates were perpendicularly embedded in the channels, and the asymmetric unit of **InP**-4

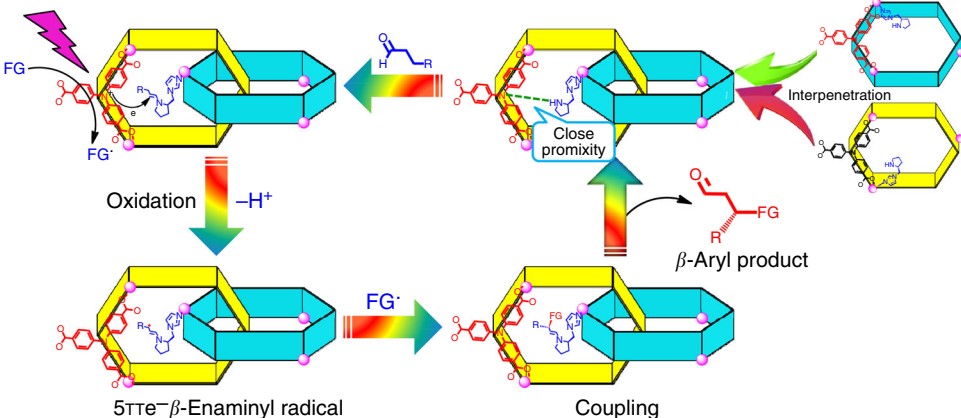

**Fig. 4** Proposed mechanisms for interpenetrated MOFs in photocatalytic β-carbonyl activation. Representation of the electron transfer pathway relevant to the photoredox and organocatalytic units via framework interpenetration for the β-functionalization of saturated aldehydes. *FG* functional group; *R* generic organic substituent

included 1.5 propionaldehyde molecules. Each substrate molecule was disordered and divided into two parts because of molecular thermal movement. One part of the carbonyl groups was located close to the **PYI** moiety, and the shortest distance between the pyrrolidine nitrogen and the carbonyl oxygen atoms of propionaldehyde was 2.97 Å. Such a close distance demonstrated a substrate—framework N–H…O hydrogen bond, which may provide an additional driving force to enforce the substrate approaching the active amine catalytic sites closely and facilitating the enamine activation. Several strong O–H…O hydrogen bond interactions (2.68 and 2.35 Å) were also observed between the other part of the carbonyl groups and the free solvent molecules (O15 and O16), which was postulated to be the driving force for the capture and stable location of the substrate within the channels of the catalyst framework.

When valeraldehyde and other more complex aldehydes were used as substrates (entries 2–6, Table 1), conversions between 69 and 84% were achieved in the **InP**-1 reaction systems, with the enantio excess reaching 52% (Supplementary Figs. 28–39 and 55–59). The insertion of the aromatic aldehyde radical coupling partners in the pores caused stronger aromatic stacking interactions and a better enantioselectivity than that of the aliphatic aldehyde product. It should be noted that these produced enantioselectivities were not found in the reported homogeneous work[18] or our control experiment (entry 1, Supplementary Table 7). These results suggested that the interpenetrated frameworks with confined chiral environments may enhance the enantioselectivity of the β-functionalization products. Interestingly, the selectivity is comparable to the best enantio excess achieved in homogeneous β-arylation reactions between 1,4-dicyanobenzene and cyclohexanone, which use chemical cocktail reaction mixtures and a larger loading (20%) of a quite complex chiral organoamine as the chiral director[18]. The simple reaction mixture in our system not only optimized the electron transfer and activation pathways to achieve efficient β-arylation transformation but also provided additional spatial constraints to enhance the enantioselectivity without introducing any additional chiral sources, which demonstrates such interpenetrated MOFs to be promising catalysts for the asymmetric photocatalytic transformation. When *L*-**PYI** was replaced by *D*-pyrrolidin-2-yl-imidazole (*D*-**PYI**), the other enantiomorph, **InP**-2 (Supplementary Data 2 and Supplementary Figs. 2 and 8), was isolated with the same cell dimensions as and a mirror-image structure to **InP**-1, exhibiting similar catalytic activity but granting products with opposite chirality under the optimum conditions (Supplementary Table 3). Interestingly, similar

substrate activation was also found in **InP**-2 systems. The valeraldehyde-impregnated **InP**-2 crystals (denoted **InP**-5) with good quality were fortunately obtained, and the single-crystal X-ray diffraction analysis suggested the maintenance of space group and main framework (Supplementary Data 5 and Supplementary Figs. 12 and 19). As depicted in Fig. 5b, the asymmetric unit of **InP**-5 contained one valeraldehyde molecule, and each substrate was perpendicularly inserted in the center of the channel. Such a special spatial occupation with the minimum steric hindrance in the channel, suggested the smooth diffusion of the substrates. The exposed active amine groups of **PYI** (N5-H5) around the inner walls of the channels provided an affinity to the carboxyl oxygen atoms of aldehyde (O17) with the N–H…O hydrogen bond (2.85 Å) interactions, which could be the potential activation forms and were extremely beneficial to the formation of enamine intermediate. Such carbonyl activation was further supported by the redshift of C=O stretching vibration from $1736 \, cm^{-1}$ (free valeraldehyde) to $1710 \, cm^{-1}$ between the IR spectra of **InP**-5 and valeraldehyde (Supplementary Fig. 17).

When the bulky aromatic aldehyde (*E*)-3-(3-oxoprop-1-enyl) phenyl 3,5-di-tert-butylbenzoate was used as a substrate[43], a <5% conversion under the same reaction condition was observed. The bulky aldehyde was too large to be adsorbed within the channels, and this size selectivity thus demonstrated that the cooperatively catalytic β-arylation reaction mainly occurred in the channels of the polymer, not on the external surface (entry 7, Table 1). To further test the impacts of external surface contact on the reaction, as-synthesized crystal samples (size: 50–80 μm) and finely ground particles (size: 0.1–1 μm) of **InP**-1 were used to catalyze the β-arylation reaction of 1,4-dicyanobenzene and benzenepropanal under the same reaction conditions (Supplementary Fig. 14). The resulting reactions gave the β-adduct with the comparable conversions of 52 and 56% after 24 h of irradiation, respectively, which demonstrated that the surface area was not the crucial factor for the transformation.

A systematic catalytic investigation was also interesting when comparing a previously reported polymer, MOF-**150**, involving **H₃NTB** as a building block[44]. As could be expected, a 5 mol% loading of MOF-**150** (per mole photosensitizer) under the same reaction conditions as the valeraldehyde system gave a negligible conversion. The excess addition of 5 mol% *L*-**PYI** to the reaction mixture led to a conversion of 52% but no *ee* value. Even with the addition increased to 20 mol%, only a 6% additional conversion and a negligible *ee* value could be detected (entries 2–4, Supplementary Table 7). The combination of **NTB³⁻** and **PYI** groups in a single framework was not only advantageous to the

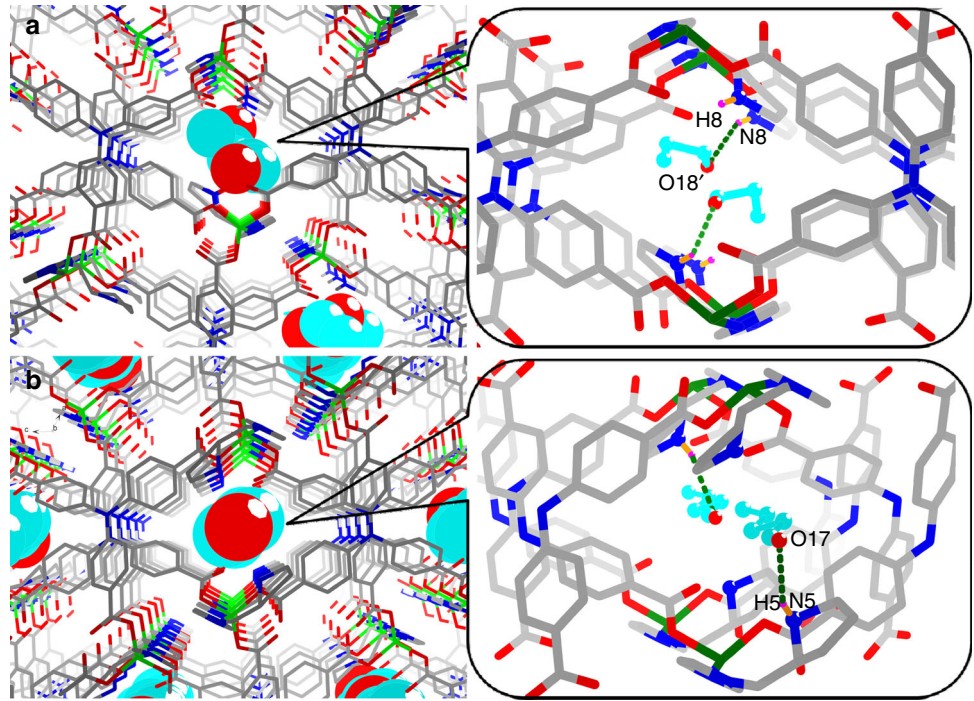

**Fig. 5** Crystal structures of photocatalysts with absorbed substrates. Single-crystal structure of **InP**-4 (**a**) and **InP**-5 (**b**), showing the capture and activation of the substrates propionaldehyde and valeraldehyde, respectively, by hydrogen-bonding interactions in the channels of the crystals

efficient energy transfer and synergetic coupling of two catalytic cycles but also to the effective collision and constrained coupling between two activated radicals. The extension of the photo-catalytic β-arylation of valeraldehyde to a previously reported non-interpenetrating Zn-**PYI**1[25], which comprised both the **NTB**$^{3-}$ and L-**PYI** groups, was also performed. With the shortest N···N separation of 10.9 Å between the pyrrolidine and triphenylamine nitrogen atoms, the loading of an idential amount of Zn-**PYI**1 under the same reaction conditions gave a conversion of 44% but negligible enantioselectivity. The modest catalytic activity and poor enantioselectivity for Zn-**PYI**1 may be caused by the longer distance between the catalytic centers in the photocatalyst and organocatalyst and the relatively uncontrolled pores generated by the stacked layers (Supplementary Fig. 13 and Supplementary Table 6). As the electron transfer from the enamine intermediate to the oxidized photoredox catalyst is the key step for the β-carbonyl activation, the superiority of an interpenetrated structure in heterogeneous systems is attributed to the close proximity between the two catalytic sites. This imposed vicinity encourages the direct interframework electron transfer for the β-activation of the carbonyl group to appear quickly and further promotes the formation of the β-enaminyl radicals via the critical 5πe$^-$ activation mode. The better enantioselectivity displayed by **InP**-1 is due to the well-modified chirality-confined pores via framework interpenetration provided more matched and restricted chiral environments to force the substrate radicals to couple with each other in a specific direction and manner.

**Photocatalytic β-functionalization of saturated ketones.** The 5πe$^-$ carbonyl activation mode was also successfully applied in the photocatalytic β-coupling of cyclic ketenes with aryl ketones by MacMillan et al. under homogeneous conditions[11]. To further assess whether our MOF catalytic systems work for such reactions, **InP**-1 was employed to facilitate the direct β-functionali-zation of saturated ketones in a heterogeneous fashion. As shown in Fig. 6, the reaction was first evaluated using cyclohexanone and

benzophenone as partners, along with 5 mol% of **InP**-1, 2 equiv of DABCO and 1 equiv of LiAsF$_6$ in DMF under visible-light irradiation. The resulting reaction gave the β-adduct with a conversion of 67% after 48 h without the formation of any α-functionalization adduct, which revealed that the 5πe$^-$ β-ketone activation was indeed applied in such a heterogeneous system (Supplementary Table 8). Control experiments suggested that no desired product was detected in the absence of catalyst or light. When 5 mol% of non-interpenetrating Zn-**PYI**1 was used for the same catalytic system, a very low conversion (<5%) of the β-adduct and fairly complicated reaction products were detected. Such a result may be ascribed to the large separation of the cat-alytic sites and the more active ketone substrates; the unsa-tisfactory separation would greatly reduce the chance of the electron being transferred from the enamine intermediate to the oxidized photosensitizer and meanwhile would create new opportunities for other species. The superiority of the inter-penetrated **InP**-1, with its closely spaced photoredox and orga-nocatalytic sites, was extended to the β-coupling of cyclohexanone analogs (Supplementary Figs. 40–54), which led to conversions from 38 to 62% under the same reaction conditions (6b–e in Fig. 6). The availability of the β-adduct in such ketone systems benefited from the smooth initiation of the photoredox and organocatalytic cycles, which were confirmed by the emission quenching experiments of benzophenone and IR studies of the activation of ketones (Supplementary Figs. 26 and 27).

## Discussion
We have reported a heterogeneous approach for the β-functio-nalization of saturated aldehydes and ketones under light irra-diation and benign environments. The approach included the incorporation of photoredox and asymmetric catalytic groups in a single framework and the interpenetration of two enantiopure frameworks to modify electron transfer between the catalytic centers. The close proximity between the photoredox sites and the

**Fig. 6** Photocatalytic β-functionalization of ketones with **InP**-1 as catalyst. Reaction conditions: benzophenone (0.5 mmol), ketones (2.5 mmol), **InP**-1 (5 mol%, based on **H₃NTB**), LiAsF₆ (0.5 mmol), DABCO (1.0 mmol), and DMF (2 mL). The conversions were determined by GC–MS analysis

asymmetric organocatalytic sites via the framework interpenetration greatly increased the chance of enamine oxidation by the oxidized photosensitizer and the formation of the key $5\pi e^-$ enaminyl radicals. The modified environments of channels provided additional spatial constraints to enhance the regioselectivity and enantioselectivity of the β-activation products without introducing any additional chiral sources. The high selectivity, good recyclability, and clear interaction pattern render the interpenetration strategy highly promising for the regulation and optimization of reaction routes in heterogeneous photocatalysis, and allow the applied protocol to be used in the β-functionalization of saturated cyclic ketones with aryl ketones to deliver γ-hydroxyketone motifs.

## Methods

**Materials and measurements**. Unless otherwise specified, all chemicals were of reagent-grade quality, were obtained from commercial sources and were used without further purification. 4,4′,4′′-Nitrilotrisbenzoicacid (**H₃NTB**)[45], L-/D-pyrrolidin-2-ylimidazole (L-/D-**PYI**)[46], MOF-**150**[44], and Zn-**PYI**1[25] were synthesized according to published procedures.

The elemental analyses of C, H, and N were performed on a Vario EL III elemental analyzer. FT-IR spectra were recorded from KBr pellets on a JASCO FT/IR-430. The powder XRD diffractograms were obtained on a Rigaku D/Max-2400 X-ray diffractometer with a sealed Cu tube ($\lambda = 1.54178$ Å). Thermogravimetric analyses were performed at a ramp rate of 10 °C/min in a nitrogen flow with an SDTQ600 instrument. Sample morphologies were recorded via scanning electron microscopy on a Jeol JSM-6390A field emission scanning electron microscope.

¹H NMR spectra were recorded on a Varian INOVA-400M type spectrometer with chemical shifts reported as ppm (TMS as the internal standard). ¹³C NMR spectra were measured on a Varian INOVA-500M spectrometer with CDCl₃ serving as an internal standard ($\delta = 77.23$). High resolution mass spectra were obtained on a GCT-CA156 Micromass GC/TOF mass spectrometer. GC analysis was performed on an Agilent Technologies 7820 A GC system. GC–MS studies were performed on a Thermo Scientific TRACE Ultra gas chromatographic instrument.

Solid UV–vis spectra were recorded on an HP 8453 spectrometer. Liquid UV–vis spectra were recorded on a TU-1900 spectrophotometer. Fluorescent spectra and lifetime measurements were recorded on an Edinburgh FS920 fluorescence spectrometer. All confocal laser scanning microscopy micrographs were collected by an Olympus Fluoview FV1000 with $\lambda_{ex} = 488$ nm.

HPLC analysis was performed on an Agilent 1100 using a ChIRALPAK AS-H column purchased from Daicel Chemical Industries, Ltd. Products were purified by flash column chromatography on 200–300 mesh silica gel and Kromasil Lc-80 using a SiO₂ chromatographic column. The CD spectra were measured on a JASCO J-810 by grinding the crystalline sample into powder to be recorded as KBr pellets.

Solid-state cyclic voltammograms were recorded on a Zahner PP211 instrument. The detailed experiments were performed with a commonly used three-electrode system (working electrode—an improved glassy carbon electrode; reference electrode—Ag/AgCl electrode; auxiliary electrode—platinum wire) in phosphate-buffered saline (PBS; scan rate: 50 mV s⁻¹; scan range: 0.4–1.2 V).

**X-ray crystallography**. Single-crystal XRD data were collected on a Bruker SMART APEX diffractometer equipped with a CCD area detector and a Mo-Kα ($\lambda = 0.71073$ Å) radiation source. The data integration and reduction were

processed using SAINT software[47]. An empirical absorption correction was applied to the collected reflections with SADABS[48]. The structures were solved by direct methods using SHELXTL and were refined on $F^2$ by the full-matrix least-squares method using the SHELXL-97 program[49, 50]. All non-hydrogen atoms in the backbone of the polymers were refined anisotropically until convergence was reached. Hydrogen atoms attached to the organic ligands were located geometrically and refined in a riding model, whereas some of the disordered solvent molecules were not treated during the structural refinements. To assist in the stability of the refinement, bond distances between several disordered atoms were fixed. Crystallographic data for **InP**-1, **InP**-2, **InP**-3, **InP**-4 and **InP**-5 are summarized in Supplementary Table 2.

**Syntheses of InP-1 and InP-2**. A mixture of **H₃NTB** (37.7 mg, 0.1 mmol), Zn(NO₃)₂· 6H₂O (29.8 mg, 0.1 mmol) and L-**PYI** (15.1 mg, 0.1 mmol) for **InP**-1 or D-**PYI** (15.1 mg, 0.1 mmol) for **InP**-2 was dissolved in mixed water (3.0 mL), CH₃OH (1.0 mL), and DMF (2.0 mL). The resulting solution was stirred for ~30 min at room temperature, sealed in a 10 mL Teflon-lined stainless steel autoclave, and heated at 110 °C for 3 days under autogenous pressure. The reaction system was then cooled to room temperature at a rate of 5 °C h⁻¹. Yellow block crystals were collected in 47% yield (based on Zn). Elemental analysis (%) Calc. for C₁₂₀H₁₂₅N₁₇O₃₁Zn₄: H, 4.92; C, 56.24; N, 9.29. Found: H, 5.12; C, 56.78; N, 8.88 for **InP**-1; Found: H, 4.98; C, 56.13; N, 8.97 for **InP**-2.

**Synthesis of InP-3**. A mixture of **H₃NTB** (37.7 mg, 0.1 mmol), Zn(NO₃)₂·6H₂O (29.8 mg, 0.1 mmol) was dissolved in mixed water (1.0 mL), pyridine (1.0 mL), and DMF (4.0 mL). The resulting solution was stirred for ~30 min at room temperature, sealed in a 10 mL Teflon-lined stainless steel autoclave, and heated at 110 °C for 3 days under autogenous pressure. The reaction system was then cooled to room temperature at a rate of 5 °C h⁻¹. Light-yellow block crystals were collected in 56% yield (based on Zn). Elemental analysis (%) Calc. for C₁₁₅H₁₁₅N₁₃O₂₉Zn₄: H, 4.82; C, 57.44.; N, 7.57. Found: H, 4.72; C, 57.89; N, 7.66.

**Synthesis of InP-4**. Crystals of **InP**-4 were obtained by soaking the crystals of **InP**-1 (0.01 mmol) in a DMF (1.5 mL) solution containing propionaldehyde (0.5 mmol) for 12 h. Elemental analysis (%) Calc. for C₁₂₅H₁₃₄N₁₆O₃₃Zn₄: H, 5.10; C, 56.65.; N, 8.46. Found: H, 4.93; C, 56.92; N, 8.36.

**Synthesis of InP-5**. Crystals of **InP**-5 were obtained by soaking the crystals of **InP**-2 (0.01 mmol) in a DMF (1.5 mL) solution containing valeraldehyde (0.5 mmol) for 12 h. Elemental analysis (%) Calc. for C₁₂₈H₁₄₀N₁₆O₃₂Zn₄: H, 5.27; C, 57.45.; N, 8.37. Found: H, 4.98; C, 57.12; N, 8.56.

**General procedure for the β-functionalization of saturated aldehydes**. A glass tube was filled with 1,4-dicyanobenzene (1.0 mmol), catalyst (0.025 mmol), and 1,4-diazabicyclo[2.2.2]octane (DABCO, 5.0 mmol). Then, this vial was purged with N₂, and the corresponding aldehyde (1.4 mmol) and DMF (3.0 mL) were added via syringe. The resulting mixture was then cooled to –78 °C, degassed and backfilled with N₂ three times. The vial was placed ~5 cm away from a 20 W fluorescent lamp source for 48 h of irradiation. The reaction mixture was extracted with ethyl acetate, dried with anhydrous Na₂SO₄, concentrated in vacuo and purified by flash chromatography on silica gel using hexane/ethyl acetate as the eluent to give the β-arylated aldehyde product as an oil. For the aromatic aldehyde substrates, the supernatant solution after centrifugation was diluted with 8 mL of DCM and 2 mL of MeOH. The reaction mixture was then cooled to 0 °C and sodium borohydride (5.0 mmol, 5.0 equiv) was added to reduce the resulting aldehyde to the

corresponding alcohol. Then the reaction mixture was treated with an identical procedure for the aliphatic aldehyde product.

**General procedure for the β-functionalization of saturated ketones**. A glass tube was filled with benzophenone (0.5 mmol), catalyst (12.5 µmol), LiAsF₆ (0.5 mmol), and DABCO (1.0 mmol). Then, this vial was purged with N₂, and the corresponding cyclohexanone (2.5 mmol, 5.0 equiv), and DMF (2.0 mL) was added via syringe. The resulting mixture was then cooled to –78 °C, degassed and backfilled with N₂ three times. The vial was placed ~5 cm away from a 20 W fluorescent lamp source for 48 h of irradiation. The reaction mixture was diluted with water and extracted with ethyl acetate. The combined organic layers were washed with brine, dried with anhydrous Na₂SO₄, concentrated in vacuo, and purified by flash chromatography on silica gel using hexane/ethyl acetate as the eluent to provide an inseparable mixture of the desired β-adduct and the corresponding hemiacetal.

**General procedure for dye-uptake measurements**. Before the adsorption of dye, **InP**-1 was first soaked in methanol solution (24 h) for guest molecule exchange and then fully dried in a vacuum oven (120 °C, 12 h) to eliminate the guest molecules (Supplementary Fig. 3). Then, the dried **InP**-1 (2.56 mg, 2 µmol) was soaked in a methanol solution of 2′,7′-dichlorofluorescein dye (24 mM, 2 mL) in a constant temperature oscillation incubator overnight. The resulting **InP**-1 was filtered and washed with methanol thoroughly until the solution became colorless and was then dried under a stream of air. The dried samples were dissociated by concentrated hydrochloric acid, and the resultant clear solution with a light olivine color was diluted to 10 mL and adjusted to a pH of 1.5. The dye concentration was determined by comparing the solution UV–vis absorption with a standard curve of the dye.

**Data availability**. The X-ray crystallographic coordinates for structures reported in this Article have been deposited at the Cambridge Crystallographic Data Centre (CCDC) under deposition numbers CCDC 1510628-1510630 and 1540231-1540232 (Supplementary Table 2). These data can be obtained free of charge from The Cambridge Crystallographic Data Centre via www.ccdc.cam.ac.uk/data_request/cif. All other data supporting the findings of this study are available within the Article and its Supplementary Information files, or from the corresponding author on reasonable request.

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

## Acknowledgements

This study was supported by the National Natural Science Foundation of China (U1608224, 21421005, and 21231003).

## Author contributions

Z.X., C.H. and C.D. conceived and designed the project. Z.X. and X.W. performed the experiments. C.H. and C.D. contributed materials and analysis tools. Z.X., C.H. and C.D. co-wrote the paper. All authors discussed the results and commented on the manuscript.

## Additional information

**Competing interests:** The authors declare no competing financial interests.

