## [Peer Review File · Nature Communications]

Reviewers' comments:

Reviewer #1 (Remarks to the Author):

The paper by Duan et al. on the β -Functionalization of Ketones and Aldehydes mediated by chiral interpenetrated MOFs is interesting and well-written. The experimental part has been properly conducted, and the authors claims supported by the collected evidence. The experimental details provided are numerous and they allow for the results to be adequately reproduced by other research groups. The different catalytic selectivity (α vs. β aldehyde/ketone derivatization) stemming from the presence of framework interpenetration is remarkable and original. The achieved results will surely be of interest to other groups working on enantioselective MOF catalysis. Overall, this is a very nice follow-up of a paper published by the same authors in 2012 (DOI:10.1021/ja305367j). It is amazing how different catalytic outcomes can be obtained by a simple synthetic procedure modification of the same heterogeneous catalyst!

For all these reasons, I strongly recommend the publication of this work on Nature Communications in its present form.

Andrea Rossin (ICCOM-CNR). ORCID ID: orcid.org/0000-0002-1283-2803.

Reviewer #2 (Remarks to the Author):

The manuscript describes the generation of a metal-organic framework (MOF) which contains both a chiral secondary amine catalyst (PYI) and an organic photocatalyst. This catalytic system was used to promote the β arylation of aldehydes.

It is in this reviewer's opinion that the study does not warrant publication in Nature Communications, since it lacks the required elements of novelty. Indeed, the concept of bringing H3NTP and PYI into close proximity by means of assembling a MOF system has been already published by the authors (reference 25) and used to promote the generation of radicals and their subsequent stereocontrolled trap by an organocatalytic chiral intermediate (in the case of Ref 25, an enamine to afford an α -functionalized aldehyde). In addition, the described chemistry is well-known, since the reported β arylation of aldehydes, driven by the generation of a radical anion and the 5n electron β -enaminyll intermediate (formed upon enamine single electron oxidation and deprotonation), was already described in 2013 using a dual photoredox organocatalysis system under homogenous conditions (reference 18). It came as a surprise when the authors have commented that their catalytic system has been "...further advanced to describe the β -functionalization of saturated cyclic ketones with aryl ketones to deliver γ -hydroxyketone motifs...2, as depicted in Table 2 of the manuscript. From this narrative it does not emerge that the very same transformation was already published using the dual photoredox organocatalysis system discussed before – it is not clear why this precedent study has been only obliquely cited as reference 14, and not clearly discussed in the text.

Besides these general aspects, there are flaws that render this manuscript unsuitable for publication in the present form. Some of the statements in the study have not been adequately corroborated by evidence, and additional studies are in need to unambiguously demonstrate some points.

The main aim of the study was to demonstrate that the intertwined metal-organic framework (MOF) containing a chiral secondary amine catalyst could bring about several advantages system in the studied beta functionalisation reactions over the dual photoredox organocatalytic system. Being the photocatalyst moiety and the secondary amine in close proximity, the electron transfer between the radical cation of the photocatalyst and the enamine should be facilitated. This is expected to lead to an improved reaction outcome and proposed to prevent side-reactions arising from the enamine reacting at the α -position with radicals. Furthermore, as the MOF generates a

chiral environment, it is suggested that higher enantioselectivities can be achieved than under standard homogeneous conditions. These claims, however, have not been proven.

- To directly compare the MOF-catalysed reaction with the standard homogeneous conditions, the transformation should be performed using the free catalysts, with the same catalyst loadings, solvent, base etc. Without this information, it cannot be decided whether or not the MOF-type catalyst leads to any improvement. For example, without knowing the enantioselectivity of the reaction with the free catalysts, it cannot be claimed that the chiral environment of the MOF leads to an increased ee of the reaction.

- It is suggested that a fast oxidation of the enamine is needed to prevent the reaction in the alpha position. The possibility of a competitive α -alkylation path, however, is rather unlikely and has never been reported in precedent studies (Refs 14 & 18). In addition, this by-product has not been detected in the present study too. In other words, there is no evidence to support that this type of reactivity really constitutes a competitive side-reaction to combat. It seems like the authors are proposing a solution for a problem that does not apply to the studied transformations (the coupling of the enamine with radicals at the alpha position).

Minor points:

- Catalyst loading: it is mentioned that 5 mol% based on the photocatalyst are used to promote the transformations. How was this catalyst loading determined?

- Figure 1 is wrong in that a methylene group is missing in most of the enamine intermediates.

- The structure of the catalyst was not clearly shown.

- About the kinetic studies, it is claimed that the reaction follows a second-order kinetic behaviour. This statement is rather obscure and should be better explained. For example, the proposed rate limiting step being the enamine oxidation would imply a zero order dependence on the dicyano benzene, and it is not clear if this is really the case.

- Concerning the supporting information and the compound characterization: in the section detailing the beta functionalisation of ketones, only one NMR spectrum is given (which is not adequately clean). For all the other compounds, only GC-MS traces of the reaction mixtures are given. NMR spectra of all the compounds should be shown instead.

In the part of the beta functionalisation of the aldehydes, the HPLC traces of some of the enantioenriched products are shown but the racemic traces are missing. They should be provided along with the missing traces for the remaining products.

Reviewer #3 (Remarks to the Author):

The manuscript by Duan and colleagues reports the heterogenization of a homogeneous multicomponent system that is able to catalyse the arylation in the beta position of alkyl carbonyl compounds (ketones and aldehydes). This is a quite a challenging reaction (see ref 18 and other work from MacMillan's group). The original homogeneous system uses a Ir(bpy)₃ as light harvester to drive the aryl coupling reaction, using the electron deficient cyano arene (R-C₆H₄-CN) substrate as the electron acceptor and the alkyl group (at the aldehyde) as the end donor. The original homogeneous system benefits from the use of an amine because it forms an adduct with the aldehyde and the resulting adduct is a good electron donor that, upon photoexcitation, forms a positive nitrogen centred radical that can further react with the activated R-C₆H₄-CN^{•-} negative radical. In this publication, the authors incorporate an amine group (pyrrolidine) in the coordination sphere at the Zn-based node of the framework. Further they changed the antenna (as an extra component not part of the framework), so that instead of the iridium complex an organic dye (fluorescein) is used. The resulting catalysts have a less broad scope of substrates under similar reaction conditions as those applied with the homogeneous catalyst. The new catalysts seemingly achieve about 5-10% of enhanced enantioselectivity compared to the original homogeneous work. This stereoselectivity is explained as a consequence of the interpenetration of the framework, however, the original homogeneous system is already stereoselective.

Overall the work seems to appeal as an "improvement" of the original work reported in reference 18 for the homogeneous system. Although the authors claim that the reaction takes place inside the porous framework, it is not clear how the bulky substrates can readily diffuse inside the pores. Perhaps the observed reactivity is only happening on the surface of the particles, as reported i.e. by Farha and coworkers in 2014 (doi: 10.1002/anie.201307520). To prove this, the authors should provide textural characterization of the catalyst before and after reaction.

Although the manuscript may become appealing for the audience of Nature Commun, there are major points to improve. Especially, written English, vague explanations and the lack of clarity in discussing the results make me advice for a rejection of the current version of the manuscript. A really important point that should be examined in detailed by the editor is the "near plagiarism" of the work from MacMillan and coworkers in reference 18. Both phrasing and specially style of figures are merely a reproduction (bad copy) of this work.

In addition to this major comments, the following points should be addressed in a revised version:

1. Lack of clarity about the challenge and major achievements of the work. In page 2, the introduction is a rewriting of the introduction of the reference 18.
2. The authors fail to explain the challenge of beta functionalization of aldehydes and give examples (ref 2-4 that are not necessarily representative).
3. In page 3 the authors claim: " we think the further modification of electron transfer between the two active sites within the frameworks via the framework interpenetration is a potential strategy for β -functionalization of saturated ketones and aldehydes (Figure 1)." But they fail to say how the H3NTB helps the system. If they incorporate fluorescein what is this amine doing as photocatalyst? The authors should provide clear evidence of their claim that the linker is able to participate in the excited state under irradiation and what is the role of the organic dye. Although they provide CV and fluorescence in the supporting information, they fail to give a clear explanation (page 6 and 7).
4. In page 3 they claim "With the closer interframework distances, the electron transfer between the two active sites would be accelerated, such that the special 5ne- β -enaminyl radical would be generated from the enamine intermediate to achieve β -functionalizational carbonyl products, before the enamine intermediate was coupled with other active radical". In my opinion the interpenetration can indeed favour the charge transfer, however I find extremely difficult to assess the real size of the adduct they claim and whether they fit in the pores of the material. Moreover, for a light driven reaction and given that they do not provide information of the life time of the separated charges, in order to react the diffusion time should be of smaller scale such that the limiting is the charge transfer (extremely fast process).
5. Page 4 "At the mean time" is an example of the language problems.
6. In page 4 in Figure 1, the scheme claim that the antenna is the H3NTB, once more, for this claim aa experiment without the organic dye fluorescein has to be done. Otherwise in my opinion the H3NTB might be only an spectator.
7. In page 5 "the interpenetration of the two frameworks enforced the close proximity between the photoredox sites and the asymmetric organocatalytic sites with the shortest interframework NN separation between the pyrrolidine and trisphenyl- amine nitrogen shortened to 6.7 Å, comparing to 8.4 Å of that in each framework of aforementioned pair (Figures 2b and 2d)." The pore diameter they refer to is for the non-interpenetrated framework. It is not clear what is exactly the point. Even in the case that the interpenetrated framework has appropriate distances, the authors do no comment on the pore volume, perhaps a more relevant concept for catalysis.
8. In page 7 they give data for the uptake of the organic dye. But then is irrelevant the distances that the author discuss in the previous page. Loading such a big molecule, the fluorescein, reduce considerably the distances.
9. In page 7 "...denser triphenylamine distribution, which were conducive to the entrancing of substrates, photon absorption and energy transport." What does this mean?
10. In page 7, the second paragraph is remarkably badly written English. I guess they try to highlight the benefits of the framework over the homogeneous system, but is only claimed but not explained.
11. In page 9 "Time dependent conversions showed that the reaction rate decreased with an increase of time, and the well-fitted linear relationship of the dynamic curve suggested a second-

ordered kinetic behavior (Figures 3c and 3d). As both the radical coupling reaction and the photo-induced electron transfer reaction are traditionally fast, the second order kinetic behaviour might be one of the proofs for that the oxidation of the enamine intermediate is the rate-limiting step to influence the whole reaction." I guess this is not a second order kinetics. This might point out to diffusion limitations. And considering that details as surface area or pore volume before and after the reaction are not given, drawing such conclusion seems rather adventurous.

12. In page 9, "Control experiments showed that no transformation occurred in the presence of only the H3NTB or PYI ligands or in the absence of any catalysts." The authors should provide an additional control experiment including the organic dye and the linkers H3NTB and PYI. Additional control experiments done with pyridine does not make sense. It will be more profitable to use imidazole without the pyrrolidine group.

13. In page 11, "The IR spectrum of InP-1 immersed in a DMF solution of propionaldehyde revealed a C=O stretching vibration at 1724 cm⁻¹. The red shift from 1736 cm⁻¹ (free propionaldehyde) suggested the adsorption and activation of propionaldehyde in the pores of the polymer, which facilitated the formation of the key enamine intermediate (Figure S14). The fact that the relevant shift was not observed in the IR spectrum of InP-3 adsorbed propionaldehyde confirmed the inactivity of InP-3 under the catalytic processes." The activation of the aldehyde with the pyrrolidene will produce an adduct that effectively is an imine. With such a small shift I am not sure you can claim that this adduct is present. It is clear that there is interaction of the framework due to the pyrrolidine but this does not provide enough information about the "key enamine intermediate".

14. In page 13, the authors discuss the activity of other non-interpenetrated frameworks. It appears that there is no stereoselectivity, however it is not clear to me why the homogeneous system affords enantioselectivity if they don't use directing agents or chiral catalysts. The authors should then provide an explanation if they want to claim the interpenetration as their argument.

15. In page 15 "The new approach included the incorporating photoredox group and asymmetric organocatalytic group in a single framework" I disagree. The asymmetric organocatalyst sits in the framework but it is not fully supported the claim for the photoredox group.

16. "The dynamic interpenetration frameworks" Why dynamic?

17. The solid state Circular Dichroism spectra. While a CD spectra of a solution always means enantiomeric excess, this is not necessarily true for solid state materials: any birefringent material, (so any material that is not cubic) might give a CD signal. Therefore you would normally rotate the sample along the azimuthal angle and additionally measure the sample flipped front to back, to exclude the artifacts from the "true" CD signal. That being said, with the ee% they reach in the conversion, I don't doubt that the samples are chiral. The experiments were simply not performed in the appropriate way. See <http://aip.scitation.org/doi/pdf/10.1063/1.1400157> and <http://onlinelibrary.wiley.com/doi/10.1002/chir.20770/epdf>

Reviewer 1's comments:

The paper by Duan *et al.* on the β -Functionalization of Ketones and Aldehydes mediated by chiral interpenetrated MOFs is interesting and well-written. The experimental part has been properly conducted, and the authors claims supported by the collected evidence. The experimental details provided are numerous and they allow for the results to be adequately reproduced by other research groups. The different catalytic selectivity (α vs. β aldehyde/ketone derivatization) stemming from the presence of framework interpenetration is remarkable and original. The achieved results will surely be of interest to other groups working on enantioselective MOF catalysis. Overall, this is a very nice follow-up of a paper published by the same authors in 2012 (DOI:10.1021/ja305367j). It is amazing how different catalytic outcomes can be obtained by a simple synthetic procedure modification of the same heterogeneous catalyst!

For all these reasons, I strongly recommend the publication of this work on *Nature Communications* in its present form.

Responses: Many thanks to the referee for the positive and kind comments.

Reviewer 2's comments:

Comment 1: The manuscript describes the generation of a metal-organic framework (MOF) which contains both a chiral secondary amine catalyst (PYI) and an organic photocatalyst. This catalytic system was used to promote the β arylation of aldehydes. It is in this reviewer's opinion that the study does not warrant publication in Nature Communications, since it lacks the required elements of novelty. Indeed, the concept of bringing H₃NTP and PYI into close proximity by means of assembling a MOF system has been already published by the authors (reference 25) and used to promote the generation of radicals and their subsequent stereocontrolled trap by an organocatalytic chiral intermediate (in the case of Ref 25, an enamine to afford an α -functionalized aldehyde). In addition, the described chemistry is well-known, since the reported β arylation of aldehydes, driven by the generation of a radical anion and the 5π electron β -enaminyll intermediate (formed upon enamine single electron oxidation and deprotonation), was already described in 2013 using a dual photoredox organocatalysis system under homogenous conditions (reference 18).

Responses: Many thanks to the referee for the constructive comments. For the photocatalytic α -functionalization of aldehyde (Ref 25), the catalytic performances and the enantioselectivity revealed that the combination of triphenylamine moiety and the PYI group within one framework was a promising approach to construct dual functional heterogeneous photocatalysts for the asymmetric transformation of carbonyls. However, this approach *via* simple integration is not powerful enough for the β -functionalization of carbonyls. As the achievement of β -functionalized product requires the quick generation of the critical $5\pi e^-$ β -enaminyll radicals from the pre-formed enamine intermediate, more precise regulation and optimization of the electron transfer pathways between the two catalytic cycles should be concerned during the design of the catalysts. In the manuscript, a new approach, that included the incorporation of both photoredox group and asymmetric organocatalytic group in a single framework, and most important, the interpenetration of the two enantiopure MOFs to modify the electron transfer pathways between the catalytic sites was developed. As exhibited in the manuscript, the new approach shortens the separations between photoredox and asymmetric catalytic sites, enhances the opportunity of the enamine oxidation to form $5\pi e^-$ β -enaminyll radical, before the enamine intermediate was coupled with other active species.

The elements of novelty of our manuscript should be mentioned and summered as: Our MOFs-based catalyst is the first example of the heterogeneous photocatalyst for the direct β -functionalization of carbonyl; The framework-interpenetration approach is powerful to modify the electron transfer pathways in the heterogeneous systems, and could be extended to other important and interesting chemical transformations. The strategy to modify the reaction environment of the pores *via* positioning multifunctional groups that accompanied by the inorganic nodes and the organic linkers of the MOFs is encouraging to enhance the regioselectivity and enantioselectivity, and at the same time make the recyclable heterogeneous catalysis occur in a manageable condition. We believe that the scientific content of this manuscript is appropriate for the audience of Nature Communications.

Of course, the chemistry of the β -arylation of aldehydes that driven by the generation of a radical anion and the 5π electron β -enaminyll intermediate, was described by using a dual photoredox metal-organic catalysis system under homogenous conditions. As the heterogenizing of homogeneous catalysts is a major endeavor in synthetic chemistry, especially for these chemical transformations using heavy transition metal compounds as catalysts, the recoverability and recyclability of our heterogeneous catalysts reported exhibit several kinds of advantages. At the meantime, the well-defined strategy for the modification of the microenvironment of the pores provides new chances to enhance the regioselectivity and the enantioselectivity of the conversions, and makes the recyclable heterogeneous catalysis occur in a manageable condition.

Comment 2: It came as a surprise when the authors have commented that their catalytic system has been "...further advanced to describe the β -functionalization of saturated cyclic ketones with aryl ketones to deliver γ -hydroxyketone motifs...2, as depicted in Table 2 of the manuscript. From this narrative it does not emerge that the very same transformation was already published using the dual photoredox organocatalysis system discussed before – it is not clear why this precedent study has been only obliquely cited as reference 14, and not clearly discussed in the text.

Responses: We are very sorry for our negligence of detail discussions in the text. According to the referee's comment, we have supplemented the discussions in the revised manuscript, "The $5\pi e^-$ carbonyl activation mode was also successfully applied in the photocatalytic β -coupling of cyclic ketenes with aryl ketones by MacMillan *et al.* under homogenous conditions (reference 14), to further assess whether our MOF catalytic systems work for such reactions, **InPs** were employed to facilitate the direct β -functionalization of saturated ketones in a heterogeneous fashion."

Comment 3: Besides these general aspects, there are flaws that render this manuscript unsuitable for publication in the present form. Some of the statements in the study have not been adequately corroborated by evidence, and additional studies are in need to unambiguously demonstrate some points. The main aim of the study was to demonstrate that the intertwined metal-organic framework (MOF) containing a chiral secondary amine catalyst could bring about several advantages system in the studied beta functionalisation reactions over the dual photoredox organocatalytic system. Being the photocatalyst moiety and the secondary amine in close proximity, the electron transfer between the radical cation of the photocatalyst and the enamine should be facilitated. This is expected to lead to an improved reaction outcome and proposed to prevent side-reactions arising from the enamine reacting at the α -position with radicals. Furthermore, as the MOF generates a chiral environment, it is suggested that higher enantioselectivities can be achieved than under standard homogeneous conditions. These claims, however, have not been proven.

Responses: Of course, these claims that were mentioned by the referee are the basic

stones supporting the novelty of our manuscript. First the competition between the α -position and β -position coupling reactions was natural existence of the photocatalytic radical coupling reactions of carbonyls. The fact that the interpenetration of frameworks caused the close proximity between the photoredox and asymmetric catalytic sites is characterized by that the shortest intra-framework N...N separations (8.4 Å) between pyrrolidine N atom and triphenylamine N atoms is 1.6 Å longer than those of inter-framework N...N separations between the two types of catalytic sites (6.8 Å). The improvement of the catalytic reaction is supported by the fact that the 5 mol% loading of the interpenetrated framework **InP**-1 gave a conversion of 84% of the β -functionalization without the formation of any α -functionalized adduct, whereas the loading of 5 mol% of Zn-**PYI**1 under the same reaction conditions gave a conversion of 44% but negligible enantioselectivity.

For the enantioselectivity of the β -arylation of aldehydes, there was no *ee* value for the β -adducts reported in the dual photoredox/organocatalytic system described by MacMillan *et al.* and we also did not detect any enantioselectivity in our control experiments (Table S4). Alternatively, our heterogeneous catalysts gave the enantioselectivity range of 29%-55% without introducing any additional chiral sources. This small but significant enantio excess should be attributed to the chiral environment of the channels that generated by the interpenetration of the frameworks, as lots of references reported by us and other research groups have demonstrated that the chiral environments of the pores within the homochiral MOFs have the ability to provide additional conformational constraints, and at the same time to enhance the enantioselectivity for the asymmetric chemical transformation.

Comment 4: To directly compare the MOF-catalyzed reaction with the standard homogenous conditions, the transformation should be performed using the free catalysts, with the same catalyst loadings, solvent, base etc. Without this information, it cannot be decided whether or not the MOF-type catalyst leads to any improvement. For example, without knowing the enantioselectivity of the reaction with the free catalysts, it cannot be claimed that the chiral environment of the MOF leads to an increased *ee* of the reaction.

Responses: We agree that the direct comparison of activities between the heterogeneous catalysis condition and the standard homogenous conditions, and the transformation should be performed using the free catalysts, with the same catalyst loadings. We are very sorry for that we did not discuss the control experiments in detail, although most of the control experiments have been performed and shown in the supporting information. The related descriptions and additional experiments have been supplemented in main text and supporting information (Table S4). In accordance with the referee's comments, the control experiments revealed that no β -arylation product was detected with the free catalysts and the same adducts and solvent as those in the standard homogenous systems.

For more details, it should be mentioned that in the original homogeneous system (Ref. 18), only one example of the another β -arylation reaction between 1,4-dicyanobenzene and cyclohexanone gave a 50% *ee* value of the product, this unique asymmetric transformation

required the addition a 20 mol% of a bulky cinchona derived chiral co-catalyst, while no enantioselectivity was reported for all the β -arylation reaction of other aldehydes. Our control experiments that the using of these essential constitutes of our catalysts did not exhibit any detectable enantio excess. Alternatively, our heterogeneous catalytic systems gave the enantioselectivity range of 29%-55% without introducing any additional chiral sources. It is postulated that the chiral environments of the channels in our interpenetrated frameworks provided additional conformational constraints to increase the enantioselectivity of these asymmetric chemical transformations.

Comment 5: It is suggested that a fast oxidation of the enamine is needed to prevent the reaction in the alpha position. The possibility of a competitive α -alkylation path, however, is rather unlikely and has never been reported in precedent studies (Refs 14 & 18). In addition, this by-product has not been detected in the present study too. In other words, there is no evidence to support that this type of reactivity really constitutes a competitive side-reaction to combat. It seems like the authors are proposing a solution for a problem that does not apply to the studied transformations (the coupling of the enamine with radicals at the alpha position).

Responses: Comparing to α -activation of carbonyls, the direct β -activation of saturated carbonyls has proven to be a more cumbersome and challenging task owing to the typically unreactive β -C(sp³)-H bonds and competitive α -substitution or other reactions. To make the β -functionalization of carbonyls efficiently, MacMillan and co-authors developed the special $5\pi e^-$ activation mode to enhance the possibility for the β -transformation, and at the same time, selected the electron-deficient 1,4-dicyanobenzene as substrate to decrease the possibility for the α -transformation (Refs 14 & 18).

In the recent manuscript, we carried on the similar substrate maps to Refs 14 & 18. As mentioned in the manuscript, the loading of 5 mol% of interpenetrated framework **InP-1** gave a conversion of 84% of the β -functionalization without the formation of any α -functionalized adduct. This result possibly demonstrated that our synthetic strategy of the heterogeneous catalysts is powerful. And at the same time, the loading of 5 mol % of **Zn-PYI1** (isolated framework that have the same constitutes as interpenetrated **InP-1**, and that was used to catalyze α -functionalization of carbonyl Ref. 25) gave 44% conversion of β -functionalized product under the same reaction conditions.

It should be also noted that our catalytic protocol could be extend to β -functionalization of the hard-to-reduce aryl ketones with cyclic ketones to deliver γ -hydroxyketone motifs. The loading of 5 mol% of our catalyst results in conversions from 38% to 67% of β -coupling of cyclohexanone and its analogues under the identical reaction conditions (Table 2). Control experiments suggested that the loading of 5 mol % of **Zn-PYI1** did not give any detectable β -functionalized products (<5%) under the same reaction conditions.

Clearly, the interpenetration of these kinds of dual functional frameworks is an important and interesting approach to controlling the efficiency of the different activation modes, and further controlling the product of the chemical transformation.

Comment 6: Catalyst loading: it is mentioned that 5 mol% based on the photocatalyst are used to promote the transformations. How was this catalyst loading determined?

Responses: The catalyst loading was determined by the mole ratio of photocatalytic active group (**H₃NTB**) to the 1,4-dicyanobenzene.

Comment 7: Figure 1 is wrong in that a methylene group is missing in most of the enamine intermediates.

Responses: The incorrect enamine intermediates in Figure 1 have been corrected.

Comment 8: The structure of the catalyst was not clearly shown.

Responses: According to the referee's comments, Figure 2 showing the structure of the catalyst has been reproduced. The asymmetric unit, the distribution of catalytic sites and framework interpenetration have also been clearly depicted.

Comment 9: About the kinetic studies, it is claimed that the reaction follows a second-order kinetic behaviour. This statement is rather obscure and should be better explained. For example, the proposed rate limiting step being the enamine oxidation would imply a zero order dependence on the dicyano benzene, and it is not clear if this is really the case.

Responses: Based on the referee's comments, we deeply explored the kinetic behavior of the heterogeneous catalytic transformations. It is true that the photocatalytic transformation is quite complicated, and it is sensitive to the effects of environments of the channels and the surface of the catalysts as well as several other complicated factors during the reaction processes. To avoid the unnecessary dispute in future, the related kinetic experimental description in the origin manuscript has been deleted in the revised version.

However, we want to give some words to explain our assumption: For the oxidization reaction, the rate-determining step that mentioned by our manuscript, the kinetic behaviors are controlled by the concentrations of the oxidized photosensitizer and enamine intermediate, as well as the collision between these two species. Generally, for the heterogeneous catalytic transition, the rate of the reaction is zero order dependence on the concentration of substrates, since the concentrations of the two species did not vary under the heterogeneous conditions. However, for the liquid/solid heterogeneous systems, the rate of the reaction should be controlled by the densities or the probability of the two species formed under the reaction conditions. In this case, a second order dependence on the dicyanobenzene and the aldehyde is assumed.

It should be noted that the quenching experiments suggested that the probability of the oxidized photosensitizer in the crystals is controlled by the Volmer equation. This means that

the concentration (or the density) of the oxidized photosensitizer in the crystalline solids, is linear related to the concentration of the quencher, the 1,4-dicyanobenzene. At the same time, as confirmed by the adsorption experiments, the higher the concentration of the aldehyde, the more enamine intermediates could be formed from the active sites in the crystalline solid catalysts. Despite these are several other factors influence the kinetic behaviour of the heterogeneous catalytic reactions, the second order dependence on the dicyanobenzene and the aldehyde, in my opinion, should be more adequate.

Comment 10: Concerning the supporting information and the compound characterization: in the section detailing the beta functionalisation of ketones, only one NMR spectrum is given (which is not adequately clean). For all the other compounds, only GC-MS traces of the reaction mixtures are given. NMR spectra of all the compounds should be shown instead.

Responses: Many thanks to the referee for the helpful suggestions. The NMR spectra of all the other products of the β -functionalization of ketones have been supplemented in the revised supporting information (Figures S44, S47, S50 and S53).

Comment 11: In the part of the beta functionalisation of the aldehydes, the HPLC traces of some of the enantioenriched products are shown but the racemic traces are missing. They should be provided along with the missing traces for the remaining products.

Responses: In accordance with the referee's comments, the relevant HPLC traces of the racemic products have been supplemented in the revised supporting information (Figures S54-58).

Reviewer 3's comments:

Comment 1: The manuscript by Duan and colleagues reports the heterogenization of a homogeneous multicomponent system that is able to catalyse the arylation in the beta position of alkyl carbonyl compounds (ketons and aldehydes). This is a quite a challenging reaction (see ref 18 and other work from MacMillan's group). The original homogeneous system uses a Ir(bipy)₃ as light harvester to drive the aryl coupling reaction, using the electron deficient cyano arene (R-C₆H₄-CN) substrate as the electron acceptor and the alkyl group (at the aldehyde) as the end donor. The original homogeneous system benefits from the use of an amine because it forms an adduct with the aldehyde and the resulting adduct is a good electron donor that, upon photoexcitation, forms a positive nitrogen centred radical that can further react with the activated R-C₆H₄-CN^{•-} negative radical. In this publication, the authors incorporate an amine group (pyrrolidine) in the coordination sphere at the Zn-based node of the framework. Further they changed the antenna (as an extra component not part of the framework), so that instead of the iridium complex an organic dye (fluorescein) is used. The resulting catalysts have a less broad scope of substrates under similar reaction conditions as those applied with the homogeneous catalyst. The new catalysts seemingly achieve about 5-10% of enhanced enantioselectivity compared to the original homogeneous work. This stereoselectivity is explained as a consequence of the interpenetration of the framework, however, the original homogeneous system is already stereoselective.

Responses: We are very sorry for our unclear statement to make the misunderstanding of referee on the properties of our photosensitizer **NTB³⁻** ligands and at the same time the dye-uptake experiment. Before the point-to-point responses to the comments, we have to give the announcement on that the fluorescein dye is only used to test the size of the channels of our catalysts, and it is never be used as the photosensitizer in the catalytic experiments.

For the repeated comments on the role of **NTB³⁻** (Comments 1, 6, 9, 11 and 18): It should be clearly noted that the **NTB³⁻** is the main component of the catalyst framework and the photosensitizer in the catalytic system. The excellent photoredox performance of **NTB³⁻** has been confirmed in our previously reported work (Ref. 25) and other groups' work. In this manuscript, the excited state of **NTB³⁻** moieties undergoes electron transfer to substrate 1,4-dicyanobenzene to afford the radical anion and the oxidation state of **NTB³⁻** moieties, which further quickly oxidize the enamine intermediate to generate the key $5\pi e^-$ β -enaminy radical and return to the ground state. Control experiments clearly demonstrated that there was no β -functionalization product to be detected without the **NTB³⁻**. We thus concluded that the photoactive **NTB³⁻** groups were really the photoredox units in the MOF-based catalysts.

It should be also noted that our catalytic system did not contain any fluorescein. The fluorescein dye mentioned in the manuscript was only used for the dye-uptake and the confocal laser scanning microscopy experiments, which evaluated the size and stability of the pores of the MOF-based catalysts even in the present of solvents. Because our catalytic reactions were performed in a liquid/solid heterogeneous system, the assessment of the characteristics of the channels of the catalysts is important and necessary. Our experiments

clearly demonstrated the bulky substrates could readily diffuse inside the pores, and suggested the MOF-based catalysts were suitable for heterogeneous catalysis. The catalysts we used were the crystals of **InPs**, they did not contain the organic dyes fluorescein.

For the comparison of the enantioselectivity between our catalytic systems and the previously reported systems, it should be mentioned in the original homogeneous work (Ref. 18) that only one example of the another β -arylation reaction between 1,4-dicyanobenzene and cyclohexanone gave a 50% *ee* value of the product, and the asymmetric transformation required the addition of a 20 mol% of a bulky cinchona-derived chiral co-catalyst, while no enantioselectivity was reported for the β -arylation reaction of other aldehydes. Despite the enantioselectivity is only 29%-55% for our heterogeneous systems, the small but significant *ee* values of the β -adducts resulting from our heterogeneous catalysts without introducing any additional chiral sources suggested that the MOF-type catalysts led to an improvement on the enantioselectivity of the products.

For the similar reaction conditions between our systems and the original homogeneous systems the referee mentioned, it is also noted that in the previous reported examples, the efficient formation of β -adducts requires quite complex reaction conditions (multiple additives: base, water, acid, photosensitizer, catalyst and DMPU solvent). Alternatively, our MOF photocatalysts facilitated the reaction only in the presence of the organic base DABCO in DMF solution, which made our reaction conditions simpler, milder and more manageable.

Moreover, we heterogenized the inexpensive organic photoredox catalyst **NTB** into the frameworks to replace the noble-metal $\text{Ir}(\text{ppy})_3$ in the homogeneous system, and the good recoverability and recyclability of the heterogeneous MOF catalysts not only eliminated the chance of contaminating organic products with trace amounts of heavy metals but also reduced the processing and waste-disposal costs in large-scale reactions, rendering our system less costly and more environmentally friendly. Therefore, the reaction conditions in this work are different from those in original homogeneous systems.

Comment 2: Overall the work seems to appeal as an “improvement” of the original work reported in reference 18 for the homogeneous system. Although the authors claim that the reaction takes place inside the porous framework, it is not clear how the bulky substrates can readily diffuse inside the pores. Perhaps the observed reactivity is only happening on the surface of the particles, as reported i.e. by Farha and coworkers in 2014 (doi: 10.1002/anie.201307520). To prove this, the authors should provide textural characterization of the catalyst before and after reaction.

Responses: We have done several experiments to confirm that the β -functionalized transformation occurred mainly inside the channels of the catalyst, not on the external surface. The experiments were summarized as follows:

First, we tested the size selectivity of substrate in this reaction. When the aliphatic aldehydes and smaller aromatic aldehydes were used as the substrates (entries 1-6, Table 1), the reactions gave a conversion range of 69%-87%. However, in the presence of the bulky

aldehyde, (*E*)-3-(3-oxoprop-1-enyl) phenyl 3,5-di-*tert*-butylbenzoate (entry 7, Table 1), whose size is larger than the pore size of **InP-1** as a substrate, the photocatalytic β -functionalization reaction only gave a less than 5% conversion under the same reaction condition, which showed that this bulky aldehyde is too large to diffuse inside the pores to interact with the pyrrolidene. The size-selectivity of the substrate suggested that the β -arylation reaction occurred mainly in the channel of the catalyst, not on the external surface.

Second, the accessible pore volume of the **InP-1** catalyst was estimated to be 18.8% of whole crystal volume by PLATON analysis, and the opening of the channels ($14.3 \text{ \AA} \times 9.4 \text{ \AA}$) of the interpenetrating MOFs is big enough for the entrance of the substrates (Table 1). Additionally, the dye-uptake studies exhibited that the **InP-1** possessed a 12.5% and 11.7% uptake of 2,7-dichlorofluorescein before and after the catalytic reaction corresponding to the framework weight, respectively (Figure S6). The substrate adsorption experiments and NMR spectra of the inclusions (Figures 3 and S19) also suggested the 1,4-dicyanobenzene and the bigger 3-(4-methoxyphenyl)propionaldehyde (entry 6, Table 1) substrates could readily diffuse inside the pores. All these results demonstrated that the accessibility of the MOFs to the substrates through the open channels.

Third, to further test the impacts of external surface contact (surface area) on the reaction, the bulk crystal sample (size: 50~80 μm) and the finely ground nanoscale particles (size: 0.1~1 μm) of **InP-1** were used to catalyze the photocatalytic β -functionalization reaction of 1,4-dicyanobenzene and benzenepropanal under the same reaction conditions (Figure S13). The resulting reactions gave the β -adduct with the comparable conversions of 52% and 56% after 24 h of irradiation, respectively, which demonstrated the surface area was not the crucial factor for the transformation.

Comment 3: Although the manuscript may become appealing for the audience of Nature Commun, there are major points to improve. Especially, written English, vague explanations and the lack of clarity in discussing the results make me advice for a rejection of the current version of the manuscript. A really important point that should be examined in detailed by the editor is the “near plagiarism” of the work from MacMillan and coworkers in reference 18. Both phrasing and specially style of figures are merely a reproduction (bad copy) of this work.

Response: After carefully considering the referee's comments, the manuscript was throughout checked and revised (marked in yellow). In the revised version of the paper, many experimental details have been extensively described and discussed to clearly explain the experimental results. A series of additional experiments (such as the homogeneous control experiments, substrate adsorption, and dye uptake and catalyst characterization) have been supplemented to prove and support the critical experimental conclusions. Moreover, most of figures in this work have been reproduced to better illustrate the content, and the presentation similar to prior homogeneous work have been modified or deleted. To improve the English,

the revised manuscript has been sent to English Language Editing Services for polishing, and we hope the new version will make the scientific content of the paper clearer for the readers.

Comment 4: Lack of clarity about the challenge and major achievements of the work. In page 2, the introduction is a rewriting of the introduction of the reference 18.

Responses: Many thanks to the referee for the comments. We rewrote the introduction part to make our major achievement and the challenge on the topic of our chemistry more clearly. Several words were also given as the responses for the comments.

For the main challenges, we think two main points should be listed: 1) The direct β -activation of saturated carbonyls was a cumbersome and challenging task owing to the unreactive β -C(sp³)-H bonds and competitive α -substitution or other reactions. Most primary β -carbonyl substitution is achieved indirectly by the nucleophilic addition of α,β -unsaturated carbonyls, despite a few examples of direct β -activation having been reported by enamine oxidation or palladium catalysis. 2) The long-life and high-energy radical intermediate for the special β -activation requires the co-existence of several kinds of catalysts and adducts to enhance the stability and to modify the electron transfer pathway for the regioselectivity controlling. Challenge goes beyond the incorporation of suitable photoredox centres and organocatalytic centres within the catalytic systems and includes the precise regulation of the electron transfer pathways between the active sites.

Faced on these challenges, two points of the major achievements should be noted: 1) We developed a new approach to modifying the electron transfer pathways in the heterogeneous systems *via* the framework interpenetration and the first example of heterogeneous photocatalyst that integrated the photocatalysis and the asymmetric organocatalysis for the direct β -functionalization of carbonyls. The enforced close proximity between two active sites *via* framework interpenetration accelerated electron transfer reaction between the oxidized photosensitizer and the enamine intermediate, enabling the generation of the $5\pi e^-$ β -enaminyll radicals and the achievement of β -functionalized carbonyl products. 2) Our strategy also allows the modification of the environment of the channels *via* positioning multifunctional groups that accompanied by the inorganic nodes and the organic linkers of the MOFs, which is encouraging to enhance the regioselectivity and enantioselectivity, and to make the recyclable heterogeneous catalysis occur in a manageable condition, eliminating the chance of contaminating organic products with trace amounts of heavy metals, and possibly reducing the processing and waste-disposal costs in large-scale reactions

Comment 5: The authors fail to explain the challenge of beta functionalization of aldehydes and give examples (ref 2-4 that are not necessarily representative).

Response: According to the referee's comments, the challenge of β -functionalization has been illustrated in the introduction of the revised manuscript and the representative references

(Ref. 9-15) have been supplemented. These words focused on the challenge of beta functionalization of aldehydes: 1) The direct β -activation of saturated carbonyls was a cumbersome and challenging task owing to the unreactive β -C(sp³)-H bonds and competitive α -substitution or other reactions. Most primary β -carbonyl substitution is still achieved indirectly by nucleophilic addition of α,β -unsaturated carbonyls, despite a few examples of direct β -activation having been reported by enamine oxidation or palladium catalysis. 2) The long-life and high-energy radical intermediate for the special β -activation requires the co-existence of several kinds of catalysts and adducts to enhance the stability and to modify the electron transfer pathways for the regioselectivity controlling.

These words have been clearly described in these References: Zhang, S. L. *et al. Nat. Commun.* **2**, 211–218 (2011); Huang, Z. & Dong, G. *J. Am. Chem. Soc.* **135**, 17747–17750 (2013); Petronijević, F. R., Nappi, M. & MacMillan, D. W. C. *J. Am. Chem. Soc.* **135**, 18323–18326 (2013). For the examples of beta functionalization of α,β -unsaturated carbonyls: Ibrahim, I., Ma, G., Afewerki, S. & Córdova, A. *Angew. Chem. Int. Ed.* **52**, 878–882 (2013); Itooka, R., Iguchi, Y. & Miyaura, N. *J. Org. Chem.* **68**, 6000–6004 (2003). For the rare examples of beta functionalization of esters and amides: Leskinen, M. V., Yip, K.-T., Valkonen, A. & Pihko, P. M. *J. Am. Chem. Soc.*, **134**, 5750–5753 (2012); Wasa, M., Engle, K. M. & Yu J.-Q. *J. Am. Chem. Soc.*, **131**, 9886–9887 (2009).

Comment 6: In page 3 the authors claim: ” we think the further modification of electron transfer between the two active sites within the frameworks via the framework interpenetration is a potential strategy for β -functionalization of saturated ketones and aldehydes (Figure 1).” But they fail to say how the H3NTB helps the system. If they incorporate fluorescein what is this amine doing as photocatalyst? The authors should provide clear evidence of their claim that the linker is able to participate in the excited state under irradiation and what is the role of the organic dye. Although they provide CV and fluorescence in the supporting information, they fail to give a clear explanation (page 6 and 7).

Response: Many thanks to the referee for the comments. We are very sorry for having the role of NTB³⁻ been misunderstood. For more details on the role of NTB³⁻ and fluorescein, please refer to the response to Comment 1.

Comment 7: In page 3 they claim “With the closer interframework distances, the electron transfer between the two active sites would be accelerated, such that the special 5π - β -enaminyl radical would be generated from the enamine intermediate to achieve β -functionalization of carbonyl products, before the enamine intermediate was coupled with other active radical”. In my opinion the interpenetration can indeed favour the charge transfer, however I find extremely difficult to assess the real size of the adduct they claim and whether they fit in the pores of the material. Moreover, for a light driven reaction and given that they do not provide information of the life time of the separated charges, in order to react the

diffusion time should be of smaller scale such that the limiting is the charge transfer.

Responses: Many thanks to the referee for the comments. There are two main photo-induced electron transfer (PET) processes for the **InPs** catalysts in the photocatalytic reaction. One is the PET from the excited state **InP*** to the substrate 1,4-dicyanobenzene, affording the arene radical anion and oxidized **InP⁺**, followed is that the oxidized state **InP⁺** oxidizes the enamine intermediate to generate the key $5\pi e^-$ β -enaminy radical anion. The former is characterized by the experimental results that the addition of 1,4-dicyanobenzene substrates to the suspension of the **InPs** catalysts, quenched the emission of the photosensitizer significantly. And the fact the lifetime of **InP-1** reduced from 4.62 ns to 3.25 ns upon addition of 1,4-dicyanobenzene (50 μ M) suggested that the PET processes is faster than most of the electron transfer in the ground state. The latter is characterized by the smooth transformation of the β -functionalized carbonyl products.

As for the fitness of the size of the adduct and the pores of the material, several words have been added in the revised manuscript. The main sentences are: the accessible pore volume of the **InP-1** catalyst was estimated to be 18.8% of whole crystal volume by PLATON analysis, and the opening of the channels (14.3 Å \times 9.4 Å) of the interpenetrating MOFs is big enough for the entrance of the substrates (Table 1), and the dye-uptake studies exhibited that the **InP-1** possessed a 12.5% and 11.7% uptake of 2,7-dichlorofluorescein before and after the catalytic reaction corresponding to the framework weight, respectively (Figure S6). The substrate adsorption experiments and NMR spectra of the inclusions (Figures 3 and S19) suggested the 1,4-dicyanobenzene and the bigger 3-(4-methoxyphenyl)propionaldehyde (entry 6, Table 1) substrates could readily diffuse inside the pores. All these results demonstrated that the accessibility of the MOFs to the substrates through the open channels.

Comment 8: Page 4 “At the mean time” is an example of the language problems.

Response: We are very sorry for our incorrect writing. The phrase “At the mean time” has been corrected to “At the meantime” and other language problems in the manuscript have also been checked and corrected one by one. To improve the English, the revised manuscript has been sent to English Language Editing Services for polishing, and we hope the new version will make the scientific content of the paper clearer for the readers.

Comment 9: In page 4 in Figure 1, the scheme claim that the antenna is the H₃NTB, once more, for this claim *ee* experiment without the organic dye fluorescein has to be done. Otherwise in my opinion the H₃NTB might be only an spectator.

Responses: We are very sorry to make the misunderstanding. As we discussed in the main text and the comments to previous comments, our catalytic systems did not contain any fluorescein, just only the catalyst and DABCO base in DMF solvent. The dye-uptake investigation that mentioned in page 7 was only used to evaluate the size and stability of the pores of the MOF-based catalysts even in the presence of solvents. The catalysts we used were

the crystals of **InPs**, they did not contain the organic dyes fluorescein. For more details on the role of **NTB**³ and fluorescein, please refer to the response to Comment 1.

Comment 10: In page 5 “the interpenetration of the two frameworks enforced the close proximity between the photoredox sites and the asymmetric organocatalytic sites with the shortest interframework NN separation between the pyrrolidine and trisphenyl- amine nitrogen shortened to 6.7 Å, comparing to 8.4 Å of that in each framework of aforementioned pair (Figures 2b and 2d).” The pore diameter they refer to is for the non-interpenetrated framework. It is not clear what is exactly the point. Even in the case that the interpenetrated framework has appropriate distances, the authors do no comment on the pore volume, perhaps a more relevant concept for catalysis.

Responses: We are very sorry for our unclear statement in the original paper. As described in the part of the structural analysis, the as-synthesized MOF catalyst was formed by the interpenetration of two enantiopure non-interpenetrating MOFs. Therefore the sentence the referee pointed means that the shortest interframework distance between the nitrogen atom from the pyrrolidine and the nitrogen atom of triphenylamine is 6.8 Å, which is shorter and more appropriate than that of intramolecular N...N separation (8.4 Å). The clearly shortening of the separation between the two active sites is the basic stones for our synthetic strategy, and is able to modify the electron transfer pathways between the photoredox and organocatalytic units for the β -functionalization of ketones and aldehydes. Also as shown in the page 7, the accessible pore volume of the interpenetrated catalyst **InP-1** was calculated to be 18.8% of whole crystal volume (approximately 1120 Å³) by PLATON analysis, and the dye-uptake and UV spectra experiments determined a quantum uptake of 12.5% of the **InP-1** weight (Figure S6). Meanwhile, the substrate adsorption experiments and NMR spectra of the inclusions (Figures 3 and S19) further confirmed the pore volume is enough for the toiless entry of substrate.

Comment 11: In page 7 they give data for the uptake of the organic dye. But then is irrelevant the distances that the author discuss in the previous page. Loading such a big molecule, the fluorescein, reduce considerably the distances.

Response: Many thanks to the referee for the comments. As we discussed in the response to the previous comments, the dye-uptake investigation mentioned in page 7 was only used to evaluate the stability of the pores in solvent and demonstrate the bulky substrates could readily diffuse inside the pores, which suggested that the MOFs-catalysts were suitable for heterogeneous catalysis. The distances discussed in the previous page are just the structural details of the catalyst **InP-1**. The catalysts we used were the crystals of **InPs**, they did not contain the organic dyes fluorescein. For more details on the role of fluorescein, please refer to the response to Comment 1.

Comment 12: In page 7 “...denser triphenylamine distribution, which were conducive to

the entrancing of substrates, photon absorption and energy transport.” What does this mean?

Responses: We are sorry for the misunderstanding. The sentence “the interpenetration of the frameworks enabled the dynamic adjustment of the channels during the reaction process and denser triphenylamine distribution, which were conducive to the entrancing of substrates, photon absorption and energy transport” has been changed as "because the **InP**-1 was evolved from two of the s non-interpenetrating frameworks *via* interpenetration, this transformation provides a chance of adjusting the pore environment and at the same time enhancing the density of triphenylamine groups around the pores."

Comment 13: In page 7, the second paragraph is remarkably badly written English. I guess they try to highlight the benefits of the framework over the homogeneous system, but is only claimed but not explained.

Responses: According to the referee’s comments, the second paragraph in page 7 was reorganized and the detailed characteristics of our reaction systems were supplemented and compared to explain the simpler reaction conditions and advantages of our heterogeneous system over the homogeneous system. As we know, the MOF-based catalysts contained both photosensitizer and asymmetric catalyst, as well as the enriched benzoate and iminium groups in the frameworks to adjust suitable Lewis-acid/base environment, our heterogenetic catalytic platform is quite simple and manageable, it need not any additional acid/base additives that have to used in the corresponding homogeneous systems, beside the advantages on the recoverability and recyclability of catalysts.

Comment 14: In page 9 “Time dependent conversions showed that the reaction rate decreased with an increase of time, and the well-fitted linear relationship of the dynamic curve suggested a second-ordered kinetic behavior (Figures 3c and 3d). As both the radical coupling reaction and the photo-induced electron transfer reaction are traditionally fast, the second order kinetic behaviour might be one of the proofs for that the oxidization of the enamine intermediate is the rate-limiting step to influence the whole reaction.” I guess this is not a second order kinetics. This might point out to diffusion limitations. And considering that details as surface area or pore volume before and after the reaction are not given, drawing such conclusion seems rather adventurous.

Responses: Based on the referee' comment, we also deeply explored the kinetic behavior of these heterogeneous catalytic transformations. It is true that the photocatalytic transformation is quite complicated, and it is sensitive to the effects of environments of the channels and the surface of the catalysts as well as several other complicated factors during the reaction processes. To avoid the unnecessary dispute in future, the related kinetic experimental description in the origin manuscript has been deleted in the revised manuscript.

However, we want to give some words to explain our assumption: For the oxidization reaction, the rate-determining step that mentioned by our manuscript, the kinetic behaviors

are controlled by the concentrations of the oxidized photosensitizer and the enamine intermediate, as well as the collision between these two species. Generally, for the heterogeneous catalytic transition, the rate of the reaction has a zero order dependence on the substrates, since the concentrations of the two species did not vary under the heterogeneous conditions. However, for the liquid/solid heterogeneous systems, the rate of the reaction should be controlled by the densities or the probability of the two species formed under the reaction conditions. In this case, a second order dependence on the dicyanobenzene and the aldehyde is assumed.

It should be noted that the luminescence quenching experiments suggested that the probability of the oxidized photosensitizer in the crystals is controlled by the Volmer equation. This means that the concentration (or the density) of the oxidized photosensitizer in the crystalline solids, is linear related to the concentration of the quencher, the 1,4-dicyanobenzene. At the same time, as confirmed by the adsorption experiments, the higher the concentration of the aldehyde, the more enamine intermediates could be formed from the active sites in the crystalline solid catalysts. Despite these are several other factors influence the kinetic behaviour of the heterogeneous catalytic reactions, the second order dependence on the dicyanobenzene and the aldehyde, in my opinion, should be more adequate.

Comment 15: “Control experiments showed that no transformation occurred in the presence of only the H₃NTB or PYI ligands or in the absence of any catalysts.” The authors should provide an additional control experiment including the organic dye and the linkers H₃NTB and PYI. Additional control experiments done with pyridine does not make sense. It will be more profitable to use imidazole without the pyrrolidine group.

Responses: Many thanks to the referee for the comments. It should be noted that our catalytic system did not include any organic dye; the H₃NTB in the MOFs was not only the bridging ligand but also the photoredox catalyst in the reaction system. In accordance with the referee’s comments, the control experiment between 1,4-dicyanobenzene and propionaldehyde with a loading of 5 mol% H₃NTB and 5 mol% PYI gave a low conversion of 11% (Table S4), because the arms of H₃NTB rotated quickly in the solvent and made the excited state quite unstable, the lifetime of the excited state is too short to activate the substrate.

Comment 16: In page 11, “The IR spectrum of InP-1 immersed in a DMF solution of propionaldehyde revealed a C=O stretching vibration at 1724 cm⁻¹. The red shift from 1736 cm⁻¹ (free propionaldehyde) suggested the adsorption and activation of propionaldehyde in the pores of the polymer, which facilitated the formation of the key enamine intermediate (Figure S14). The fact that the relevant shift was not observed in the IR spectrum of InP-3 adsorbed propionaldehyde confirmed the inactivity of InP-3 under the catalytic processes.” The activation of the aldehyde with the pyrrolidene will produce an adduct that effectively is an imine. With such a small shift I am not sure you can claim that this adduct is present. It is

clear that there is interaction of the framework due to the pyrrolidine but this does not provide enough information about the “key enamine intermediate”.

Responses: As we described in the manuscript, the redshift of the C=O stretching vibration indicated an interaction between the aldehyde and pyrrolidine, and meant that the pyrrolidene effectively activated the aldehyde, which is beneficial to improve the probability of the formation of the enamine intermediate. We tried our best to obtain the suitable crystals of **InPs** for single-crystal X-ray diffraction analysis to explore the information of the enamine intermediate during the reaction process. In the case of the substrate-uptake experiments, the crystal structures of the **InP-1** encapsulating propionaldehyde and the **InP-2** encapsulating valeraldehyde (denoted **InP-4** and **InP-5**, respectively) were determined. As shown in Figure 5, the propionaldehyde substrates were perpendicularly embedded in the channels. The carbonyl groups were located close to the **PYI** moiety, and the shortest distance between the pyrrolidine nitrogen and the carbonyl oxygen atoms of propionaldehyde was 2.97 Å. Such a close distance demonstrated a substrate–framework N-H...O hydrogen bond, which may provide an additional driving force to enforce the substrate approaching the active amine catalytic sites closely and facilitating the enamine activation. A similar situation was also found in **InP-5**, the exposed active amine groups of **PYI** (N5-H5) around the inner walls of the channels provided an affinity to the carboxyl oxygen atoms of aldehyde (O17) with the N-H...O hydrogen bond (2.85 Å) interactions, which could be the potential activation forms and were extremely beneficial to the formation of enamine intermediate.

On the other hand, the enamine activation between carbonyls and pyrrolidines has developed into a kind of classic catalytic approach in organic synthesis over the last two decades, and well-known in Aldol, Mannich, Michael, Diels-Alder reactions *et al.*. A tremendous amount of excellent research has been directed towards identifying the mechanism involving the formation of a reactive enamine intermediate in general chemical synthesis¹⁻⁵ and in photocatalytic transformation.⁶⁻⁸

References:

1. S. Mukherjee, J. W. Yang, S. Hoffmann, and B. List. Asymmetric enamine catalysis. *Chem. Rev.* 2007, 107, 5471–5569.
2. D. W. C. MacMillan. The advent and development of organocatalysis. *Nature* 2008, 455, 304–308.
3. W. Notz, F. Tanaka, and C. F. Barbas. Enamine-based organocatalysis with proline and diamines: the development of direct catalytic asymmetric aldol, mannich, michael, and diels-alder reactions. *Acc. Chem. Res.* 2004, 37, 580–591.
4. B. List. Proline-catalyzed asymmetric reactions. *Tetrahedron* 2002, 58, 5573–5590.
5. P. Melchiorre, M. Marigo, A. Carlone, and G. Bartoli. Asymmetric aminocatalysis—gold rush in organic chemistry. *Angew. Chem. Int. Ed.* 2008, 47, 6138–6171.
6. C. K. Prier, D. A. Rankic, and D. W. C. MacMillan. Visible light photoredox catalysis with

transition metal complexes: applications in organic synthesis. *Chemical Reviews*, 2013, 113, 5322–5363.

7. D. A. Nagib, M. E. Scott, and D. W. C. MacMillan. Enantioselective α -trifluoromethylation of aldehydes *via* photoredox organocatalysis. *J. Am. Chem. Soc.* 2009, 131, 10875–10877.
8. D. A. Nicewicz and D. W. C. MacMillan. Merging photoredox catalysis with organocatalysis: the direct asymmetric alkylation of aldehydes. *Science*, 2008, 322, 77–80.

Comment 17: In page 13, the authors discuss the activity of other non-interpenetrated frameworks. It appears that there is no stereoselectivity, however it is not clear to me why the homogeneous system affords enantioselectivity if they don't use directing agents or chiral catalysts. The authors should then provide an explanation if they want to claim the interpenetration as their argument.

Response: Many thanks to the referee for the comments. As described in the homogeneous system (ref. 18), a 50% *ee* value of the β -arylation reaction between 1,4-dicyanobenzene and cyclohexanone was only observed when 20 mol% of a bulky cinchona-derived chiral cocatalyst was used in the reaction mixture, and no enantioselectivity was observed for β -arylation of aldehydes without the presence of bulky chiral agents. On the other hand, our control experiments demonstrated that the chiral pyrrolidine molecules in the homogeneous system hardly restricted the coupling patterns of the substrates and directed the chirality of the products. Only in our heterogeneous catalytic system, the asymmetric pyrrolidines were fixed on surfaces of the well-modified pores of the catalysts. These special chiral environments provide special spatial constraints to facilitate the asymmetrical β -arylation of aldehydes without the addition of any other chiral adducts.

Generally, the achievement of enantioselective products in homogeneous transformation requires the assistance of the bulky chiral directing agents with chiral steric hindrance for the conformation inhibition of substrates in an asymmetrical fashion. As for the β -arylation of saturated carbonyls, the conformation of the coupling intermediate between the $5\pi e^-$ enaminyll radical and the arene radical drove the enantioselectivity of the products. In the case of the interpenetrated **InP-1** catalyst, the well-modified chiral environment around the pores provided additional conformation constraints to restrict the absolute conformation of the intermediate, affording enantioselective products. While the chiral pores within the isolated photoactive frameworks also provide special conformation constraints during the intermediate transformation, the larger intramolecular separation between the photosensitizer and the enamine intermediate makes the generation of $5\pi e^-$ β -enaminyll radical quite difficult before the intermediate coupled with other active species, very little of β -functionalized carbonyl products was formed and no detectable enantio excess could be observed.

Comment 18: In page 15 “The new approach included the incorporating photoredox group and asymmetric organocatalytic group in a single framework” I disagree. The

asymmetric organocatalyst sits in the framework but it is not fully supported the claim for the photoredox.

Responses: Many thanks to the referee for the comments. As we discussed in the response to the previous comments, the photoactive groups **NTB**³⁻ were really the photoredox units in the MOF-based catalysts. For more details on the role of **H₃NTB**, please refer to the response to Comment 1.

Comment 19: “The dynamic interpenetration frameworks” Why dynamic?

Responses: Because the interpenetrating **InP-1** was assembled by identical frameworks with non-covalent bond interactions, the two parts in solution may give a locally appropriate adjustment within the interpenetrated framework to adapt the diffusion of the substrate or other changes such as the microenvironment of the pores of the materials.^{1,2} To avoid the unnecessary dispute in future, the related description of “dynamic” in the origin manuscript has been deleted in the revised manuscript.

The similar phenomena have also been reported in these references:

1. Karen L. Mulfort, Omar K. Farha, Christos D. Malliakas, Mercouri G. Kanatzidis, and Joseph T. Hupp. *Chem. Eur. J.* 2010, 16, 276–281.
2. Yue Wu, Vanessa K. Peterson, Emily Luks, Tamim A. Darwish, and Cameron J. Kepert. *Angew. Chem. Int. Ed.* 2014, 53, 5175–5178.

Comment 20: The solid state Circular Dichroism spectra. While a CD spectra of a solution always means enantiomeric excess, this is not necessarily true for solid state materials: any birefringent material (so any material that is not cubic) might give a CD signal. Therefore you would normally rotate the sample along the azimuthal angle and additionally measure the sample flipped front to back, to exclude the artifacts from the “true” CD signal. That being said, with the ee% they reach in the conversion, I don’t doubt that the samples are chiral. The experiments were simply not performed in the appropriate way.

Responses: Many thanks to the referee for the kind suggestions. In accordance with the method literature reported and the referee suggested, the CD spectra of six samples (three **InP-1** and three **InP-2**) with KBr pellets were measured on JASCO J-810. According to the each three test results (Table S1), we took an average value with a small standard deviation and found that there are no significant differences in the CD signals from them and from the samples we detected before.

REVIEWERS' COMMENTS:

Reviewer #2 (Remarks to the Author):

It is in the opinion of this reviewer that the revised version of the manuscript has not addressed the main concerns and the flaws identified in the first round of evaluation.

The main issue is about the central claim of this study: the authors claim that "since the achievement of the beta-functionalized product requires the quick generation of the critical 5 π - β -enaminyl radicals from the pre-formed enamine intermediate, more precise regulation and optimization of the electron transfer pathways between the two catalytic cycles should be concerned during the design of the catalysts".

They go further by saying that "the major challenge includes the precise regulation of the electron transfer pathway between the active sites".

This reviewer believes that these claims have not been adequately corroborated: The critical issue for achieving regioselective beta-functionalization of saturated carbonyls is not about the "the precise regulation or the modification of the electron transfer pathway", as stated by the authors. As clearly demonstrated by MacMillan using a homogeneous dual catalytic system, the regioselectivity arises from the unique properties of the enaminyl radical cation, which is generated upon simple SET oxidation of the transiently formed electron-rich enamine. The key mechanistic consideration is that the enaminyl radical cation sufficiently weakens the allylic C-H bonds (at the original carbonyl β -position of the carbonyl) so as to allow deprotonation and formation of the β -enamine radical, which represents the critical 5 π e-activation mode.

This pathway has nothing to do with the precise regulation of the electron transfer pathway. It requires a simple SET oxidation of an electron-rich compound and it mainly arises from the increased acidity of the resulting oxidized intermediate – homogeneous or heterogeneous approaches likely follow the same mechanism framework, and neither the control nor the modification of the electron transfer pathways between the catalytic sites are required for success. It appears instead that a fast deprotonation of the enaminyl radical cation is needed.

This means that the main claim of the paper, i.e. the interpenetration of the two enantiopure MOFs modifies the electron transfer pathways between the catalytic sites, has not been proven.

Another key statement that was not adequately proven is the following: "the competition between the α -position and β -position coupling reactions was natural existence of the photocatalytic radical coupling reactions of carbonyls".

As already mentioned in the previous evaluation, the possibility of a competitive α -alkylation path is rather unlikely and has never been reported in precedent studies. In addition, this by-product has not been detected in the present study too. In other words, there is no evidence to support that this type of reactivity really constitutes a competitive side-reaction to combat. It seems like the authors are proposing a solution for a problem that does not apply to the studied transformations (the coupling of the enamine with radicals at the alpha position).

To sum up, it is in this reviewer's opinion that the study does not warrant publication in Nature Communications. The authors mention that a crucial point is the fast generation of the 5- π intermediate and that this can only be achieved by bringing the photocatalyst in close proximity to the secondary amine moiety. However, this reaction works well under homogeneous conditions too, in which the two sites are on separate compounds. It is therefore not clear whether or not the given explanation is valid.

Furthermore, the authors mention that "the framework-interpenetration approach is powerful to modify the electron transfer pathways..." This implies that, when using the MOF-type catalyst, the mechanism is being changed, which is most likely not the case.

Reviewer #3 (Remarks to the Author):

I am very satisfied with the effort that the authors have put in improving the quality and clarity of their work. In my opinion, the manuscript is now ready for acceptance.

Response to the comments of the referees

Reviewer 2' comments

Comment 1: It is in the opinion of this reviewer that the revised version of the manuscript has not addressed the main concerns and the flaws identified in the first round of evaluation. The main issue is about the central claim of this study: the authors claim that “since the achievement of the beta-functionalized product requires the quick generation of the critical 5π - β -enaminy radical from the pre-formed enamine intermediate, more precise regulation and optimization of the electron transfer pathways between the two catalytic cycles should be concerned during the design of the catalysts”. They go further by saying that “the major challenge includes the precise regulation of the electron transfer pathway between the active sites”. This reviewer believes that these claims have not been adequately corroborated: The critical issue for achieving regioselective beta-functionalization of saturated carbonyls is not about the “the precise regulation or the modification of the electron transfer pathway”, as stated by the authors. As clearly demonstrated by MacMillan using a homogeneous dual catalytic system, the regioselectivity arises from the unique properties of the enaminy radical cation, which is generated upon simple SET oxidation of the transiently formed electron-rich enamine. The key mechanistic consideration is that the enaminy radical cation sufficiently weakens the allylic C–H bonds (at the original carbonyl β -position of the carbonyl) so as to allow deprotonation and formation of the β -enamine radical, which represents the critical 5π e^- activation mode. This pathway has nothing to do with the precise regulation of the electron transfer pathway. It requires a simple SET oxidation of an electron-rich compound and it mainly arises from the increased acidity of the resulting oxidized intermediate – homogeneous or heterogeneous approaches likely follow the same mechanism framework, and neither the control nor the modification of the electron transfer pathways between the catalytic sites are required for success. It appears instead that a fast deprotonation of the enaminy radical cation is needed. This means that the main claim of the paper, i.e. the interpenetration of the two enantiopure MOFs modifies the electron transfer pathways between the catalytic sites, has not been proven.

Responses: We are afraid that the reviewer takes a one-sided view to our statement, and ignores that our heterogeneous MOF-catalysis systems are quite different from those homogeneous systems described by the reviewer and MacMillan. In the homogeneous systems, both the photosensitizers and organocatalysts were allowed to be moved freely. They

can rapidly adjust their own location and situation during the arbitrary collision to provide the required angle, orientation and energy for the electron transformation and radical-radical coupling between substrates. However, in our heterogeneous catalysts, both the photosensitizers and organocatalysts are fixed on the framework and the produced oxidation state of photosensitizer and enamine intermediate are also immovable, it is difficult to achieve the β -functionalization product if the electron cannot be effectively transferred from the oxidized photosensitizer to the enamine intermediate to produce the key enaminy radical cation. On the other hand, the enamine intermediates are also short-lived, its rapid electron capture and oxidation are the precondition for the formation of $5\pi e^-$ β -enaminy radicals. If this transfer process is too slow, it will greatly increase the opportunity of the formation of other by-products because the enamine intermediates can be coupled with other active species.

Apparently, the precise regulation and optimization of the rapid electron transfer between the two catalytic cycles should be given important consideration during the design of the catalysts, and shortening the separations between photoredox and organocatalytic sites is exactly an effective way for improving the electron transfer. If the transfer distance is too far, the chance of successful electron transfer will be significantly reduced, which might directly lead to inefficient formation of $5\pi e^-$ β -enaminy radicals. At the same time the arene radical anion will be more likely to couple with other active species to yield by-products before coupling with the enamine radicals. To prove it, two MOF catalysts **InP-1** and **Zn-PYI1** with the same photosensitizer and organocatalyst but with different separations between two active sites (6.8 Å for **InP-1** and 10.9 Å for **Zn-PYI1**), were employed to facilitate the direct β -functionalization of saturated carbonyls and test the effects of the separation on the catalysis in our manuscript. Obviously, the yields for **InP-1** (84% for valeraldehyde and 67% for cyclohexanone) are much higher than those for **Zn-PYI1** (44% for valeraldehyde and <5% for cyclohexanone), while the corresponding by-products are much less.

In addition, with respect to the claims “It appears instead that a fast deprotonation of the enaminy radical cation is needed” mentioned by the reviewer, we do not agree with it. First, the increased acidity of the allylic C–H bonds and easy deprotonation of the β -carbonyl position of enaminy radical cation have been demonstrated experimentally and theoretically by MacMillan *et al.* in their work (*Science*, 2013, 339, 1593-1596), and the alkaline environment (5 equiv base) in system is beneficial for the fast deprotonation of the enaminy radical cation. Secondly, the formation of the enaminy radical cation is the prerequisite to achieve the $5\pi e^-$ β -enaminy radicals and β -functionalization products, and the rapid enamine

oxidization is the key step to guarantee the successful formation. Therefore, the precisely regulating the electron transfer between the active sites to increase the opportunity of successful enamine oxidization is more important.

Comment 2: Another key statement that was not adequately proven is the following: “the competition between the α -position and β -position coupling reactions was natural existence of the photocatalytic radical coupling reactions of carbonyls”. As already mentioned in the previous evaluation, the possibility of a competitive α -alkylation path is rather unlikely and has never been reported in precedent studies. In addition, this by-product has not been detected in the present study too. In other words, there is no evidence to support that this type of reactivity really constitutes a competitive side-reaction to combat. It seems like the authors are proposing a solution for a problem that does not apply to the studied transformations (the coupling of the enamine with radicals at the alpha position).

Responses: The well-known mechanism (*Chem. Rev.* 2007, 107, 5471–5569) illustrates that the formation of enamine must undergo a pre-formation of allylamine cation during the condensation of aldehyde and amine, which is apt to be attacked by the radical at the α -carbonyl position to produce the corresponding α -alkylation product. Meanwhile, the transiently formed enamine needs a rapid oxidation to form corresponding β -enaminy radical cation, or else it might be coupled with other active species to produce a variety of by-products, such as the α -carbonyl product. So the competition between the α -position and β -position coupling reactions was natural existence of the photocatalytic radical coupling reactions of carbonyls.

As mentioned in MacMillan’s homogeneous system (*Science*, 2013, 339, 1593-1596), great efforts have been paid to modify suitable environment to avoid direct reaction with enamine at the α -carbonyl position. Both the unique chemical cocktail additives and the well-chosen electron-deficient 1,4-dicyanobenzene substrate suggested the difficulty on the achievement of β -functionalization products. While honoring and appreciating these exciting contributions on the homogeneous system, we should also understand the apparent difference in heterogeneous MOF-catalysis systems due to the fixed catalytic sites. Though we carried on the similar substrate maps to Ref. 18 to decrease the possibility for the α -transformation, the influences of the separations between the two catalytic sites on the enamine oxidation to form the critical $5\pi e^-$ β -enaminy radicals and on the avoiding the α -alkylation attack cannot be overlooked. To test it, the interpenetrated **InP-1** and non-interpenetrating **Zn-PYI1** were used to facilitate the photocatalytic β -alkylation reactions between 1,4-dicyanobenzene and

propionaldehyde. After several days of separation, purification and detection, the ^1H NMR and GC-MS data (Figs 1 and 2) revealed that there are about 20% α -arylation product (α -carbonyl : β -carbonyl products = 1:4) in the reaction system of non-interpenetrating **Zn-PYI1**, while no α -adducts was detected in our interpenetrated **InP-1** system. This result demonstrated that the special heterogeneous systems allowed more loopholes of α -carbonyl attacks than those in well-modified homogeneous systems and our interpenetration strategy of the heterogeneous catalysts is powerful to avoid the α -transformation. The interpenetration of these kinds of dual functional frameworks is an important and interesting approach to regulating and modifying the electron transfer pathways, and further controlling the product of chemical transformation.

Figure 1. The ^1H NMR spectrum of the purified product generated by the catalysis of **Zn-PYI1**

Figure 2. GC-MS of the purified product generated by the catalysis of Zn-PYII

Comment 3: To sum up, it is in this reviewer’s opinion that the study does not warrant publication in Nature Communications. The authors mention that a crucial point is the fast generation of the 5-Pi intermediate and that this can only be achieved by bringing the photocatalyst in close proximity to the secondary amine moiety. However, this reaction works well under homogeneous conditions too, in which the two sites are on separate compounds. It is therefore not clear whether or not the given explanation is valid. Furthermore, the authors mention that “the framework-interpenetration approach is powerful to modify the electron transfer pathways...” This implies that, when using the MOF-type catalyst, the mechanism is being changed, which is most likely not the case.

Responses: As we know, the photosensitizers and organocatalysts in heterogeneous systems are stationary. The separation of the enamine intermediate and the oxidized photosensitizers is the important factor to form the $5\pi e^-$ β -enaminyll radicals. The interpenetration of frameworks that effectively shortened the proximity between these two active sites suggested that it should be the powerful approach for the photocatalytic β -alkylation. The details see the responses to the Comments 1 and 2.

Reviewer 3’ comment

Comment 1: I am very satisfied with the effort that the authors have put in improving the quality and clarity of their work. In my opinion, the manuscript is now ready for acceptance.

Response: Many thanks to the referee for the positive and kind comments.

Reviewer Comments:

Reviewer #2 (Remarks to the Author):

The authors have provided evidence in support of their statements that the distance between the catalytic units in their MOF-type catalyst is essential for reactivity. Those experiments greatly reinforced some of the key statements.

In the eyes of this referee, though, the main problem with this paper is that the authors make general claims that are true only in the specific settings of heterogeneous MOF-catalysis systems. For example, starting the manuscript with the following sentence: "Modifying electron transfer pathways is essential to controlling the regioselectivity of photochemical transformations relevant to saturated carbonyls" is misleading. This statement holds true only in the case studied by the authors, in which a heterogeneous MOF-catalysis system is used. In the homogeneous system (Ref 18), the control of the regioselectivity of photochemical transformations relevant to saturated carbonyls is not an issue at all.

The impression is that the authors composed their narrative in such a way to instill the perception that they are solving a general problem connected with regioselectivity in carbonyl functionalization. In reality, this issue is intrinsic to the use of heterogeneous MOF-catalysts and the difficulties related to the fixed catalytic sites. Solving this issue maybe relevant to the heterogeneous catalysis community, and it is up to the Editor to eventually decide on the suitability of the manuscript for publication. This reviewer, however, believes that the manuscript, in the present form, does not convey the real message of the paper - the precise regulation and optimization of the rapid electron transfer between the two catalytic cycles is important ONLY for heterogeneous MOF-based catalysis, since the homogeneous system is exempted from this problem.

As a note, the authors claim in their rebuttal that "As mentioned in MacMillan's homogeneous system (Science, 2013, 339, 1593-1596), great efforts have been paid to modify suitable environment to avoid direct reaction with enamine at the α -carbonyl position".

This is not true - in the original study, it is clearly specified that the reaction proceeds "without formation of any alpha-amine arylation adducts".

The manuscript should be extensively reworded before of possible publication to avoid unsubstantiated general claims. Efforts should be made to better define that this study provides a solution to the difficulties related to the fixed catalytic sites within heterogeneous MOF-based catalysts.